# Selection of Ideal Reference Genes for Gene Expression Analysis in COVID-19 and Mucormycosis

Sunil Kumar,[a] Ayaan Ahmad,[a] Namrata Kushwaha,[a] Niti Shokeen,[a] Sheetal Negi,[a] Kamini Gautam,[a] Anup Singh,[b] Pavan Tiwari,[c] Rakesh Garg,[d] Richa Agarwal,[d] Anant Mohan,[c] Anjan Trikha,[d] Alok Thakar,[b] Vikram Saini[a,e]

[a]Laboratory of Infection Biology and Translational Research, Department of Biotechnology, All India Institute of Medical Sciences, New Delhi, India
[b]Department of Otorhinolaryngology-Head & Neck Surgery, All India Institute of Medical Sciences (AIIMS), New Delhi, India
[c]Department of Pulmonary Medicine and Sleep Disorders, All India Institute of Medical Sciences (AIIMS), New Delhi, India
[d]Department of Onco-Anesthesiology, Intensive Care, Pain and Palliative Medicine, All India Institute of Medical Sciences (AIIMS), New Delhi, India
[e]Biosafety Laboratory-3, Centralized Core Research Facility (CCRF), All India Institute of Medical Sciences (AIIMS), New Delhi, India

**ABSTRACT** Selection of reference genes during real-time quantitative PCR (qRT-PCR) is critical to determine accurate and reliable mRNA expression. Nonetheless, not a single study has investigated the expression stability of candidate reference genes to determine their suitability as internal controls in SARS-CoV-2 infection or COVID-19-associated mucormycosis (CAM). Using qRT-PCR, we determined expression stability of the nine most commonly used housekeeping genes, namely, TATA-box binding protein (*TBP*), cyclophilin (*CypA*), $\beta$-2-microglobulin (*B2M*), 18S rRNA (*18S*), peroxisome proliferator-activated receptor gamma (PPARG) coactivator 1 alpha (*PGC-1$\alpha$*), glucuronidase beta (*GUSB*), hypoxanthine phosphoribosyltransferase 1 (*HPRT-1*), $\beta$-*ACTIN*, and glyceraldehyde-3-phosphate dehydrogenase (*GAPDH*) in patients with COVID-19 of various severities (asymptomatic, mild, moderate, and severe) and those with CAM. We used statistical algorithms (delta-$C_T$ [threshold cycle], NormFinder, BestKeeper, GeNorm, and RefFinder) to select the most appropriate reference gene and observed that clinical severity profoundly influences expression stability of reference genes. *CypA* demonstrated the most consistent expression irrespective of disease severity and emerged as the most suitable reference gene in COVID-19 and CAM. Incidentally, *GAPDH*, the most commonly used reference gene, showed the maximum variations in expression and emerged as the least suitable. Next, we determined expression of nuclear factor erythroid 2-related factor 2 (*NRF2*), interleukin-6 (*IL-6*), and *IL-15* using *CypA* and *GAPDH* as internal controls and show that *CypA*-normalized expression matches well with the RNA sequencing-based expression of these genes. Further, *IL-6* expression correlated well with the plasma levels of IL-6 and C-reactive protein, a marker of inflammation. In conclusion, *GAPDH* emerged as the least suitable and *CypA* as the most suitable reference gene in COVID-19 and CAM. The results highlight the expression variability of housekeeping genes due to disease severity and provide a strong rationale for identification of appropriate reference genes in other chronic conditions as well.

**IMPORTANCE** Gene expression studies are critical to develop new diagnostics, therapeutics, and prognostic modalities. However, accurate determination of expression requires data normalization with a reference gene, whose expression does not vary across different disease stages. Misidentification of a reference gene can produce inaccurate results. Unfortunately, despite the global impact of COVID-19 and an urgent unmet need for better treatment, not a single study has investigated the expression stability of housekeeping genes across the disease spectrum to determine their suitability as internal controls. Our study identifies *CypA* and then *TBP* as the two most suitable reference genes for COVID-19 and CAM. Further, *GAPDH*, the most commonly used reference gene in COVID-19 studies, turned out to be the least suitable. This work fills an important gap in the field and promises to facilitate determination of an accurate expression of genes to

Address correspondence to Vikram Saini, vikram@aiims.edu.

The authors declare no conflict of interest.

catalyze development of novel molecular diagnostics and therapeutics for improved patient care.

**KEYWORDS** gene expression, qRT-PCR, reference gene, SARS-CoV-2, COVID-19, mucormycosis, *CypA*, *NRF2*, *IL-6*

Understanding changes in gene expression is critical to develop diagnostics, therapeutics, and prognostic approaches against infectious agents (1–6). In this context, real-time quantitative PCR (qRT-PCR) is often preferred to monitor gene expression due to its high sensitivity and specificity and broad quantification range. However, for qRT-PCR an endogenous control or reference gene from the same sample is essential for an accurate assessment of gene expression. Not only must a suitable reference gene have an adequate and consistent expression, but more importantly, the reference gene should show minimal expression variability between samples and under experimental conditions. Nonetheless, the reference genes may often demonstrate significant variability in expression, compromising outcomes and even producing inaccurate results (7–9). Therefore, validation of reference genes in any new experimental system is of utmost importance.

Housekeeping genes, due to their constitutive and stable expression, often qualify as reference genes in qRT-PCR studies (10–12). Indeed, housekeeping genes like glyceraldehyde-3-phosphate dehydrogenase (*GAPDH*), hypoxanthine guanine phosphoribosyl transferase 1 (*HPRT-1*), cyclophilin A (*CypA*), ribosomal protein L13A (*RPL13A*), and *β-ACTIN* are among the most commonly used reference genes or internal controls for qRT-PCR analysis (13, 14). For viral infections including COVID-19, *β-ACTIN*, *GAPDH*, and 18S rRNA (18S) are popularly used as internal controls (4, 15–18). However, most of these genes do not consistently manifest stable expression under various experimental conditions (19–21). This is critical during infections like those of SARS-CoV-2 that manifest a spectrum of disease ranging from asymptomatic through mild and moderate to severe, each with distinct clinical features (22–24). Moreover, onset of acute coinfections, especially those with fungi (mucormycosis), during SARS-CoV-2 infection or following recovery may further impact the stability and expression of the reference genes. Notably, over 45,000 people in India have developed COVID-19-associated mucormycosis (CAM), which has a fatality rate of >50% and was declared an epidemic in India (25). Unfortunately, despite the high mortality of CAM and the ongoing COVID-19 pandemic, there is not a single study focused on identifying the appropriate reference genes and investigating the impact of COVID-19 or CAM on commonly used reference genes.

In this study, we have addressed these knowledge gaps by evaluating the gene expression stability of nine widely used housekeeping genes in peripheral blood mononuclear cells (PBMCs) isolated from patients having a spectrum of COVID-19 and patients who developed CAM. Our results clearly show significant variations in the expression of housekeeping genes during SARS-CoV-2 infection or CAM, and identify the most appropriate reference gene establishing the significance of the study.

## RESULTS

**Melt curve analyses indicate specificity of the gene primers.** From clinically confirmed cases of COVID-19 and CAM, we isolated PBMCs, obtained good-quality RNA, and performed cDNA synthesis. All primer pairs used in the study yielded the best amplification efficiency at an annealing temperature of 60°C (see Fig. S1 in the supplemental material). We observed a single amplification product for each primer pair as evident from the presence of a single peak in the melt curve analysis (Fig. 1). This highlighted the absence of nonspecific binding of primer pairs, justifying their suitability for further downstream applications.

**Analysis of candidate reference genes reveals heterogeneity in gene expression based on disease severity in COVID-19 and CAM patients.** We next investigated the changes in the expression of candidate housekeeping genes in patients ($n = 56$)

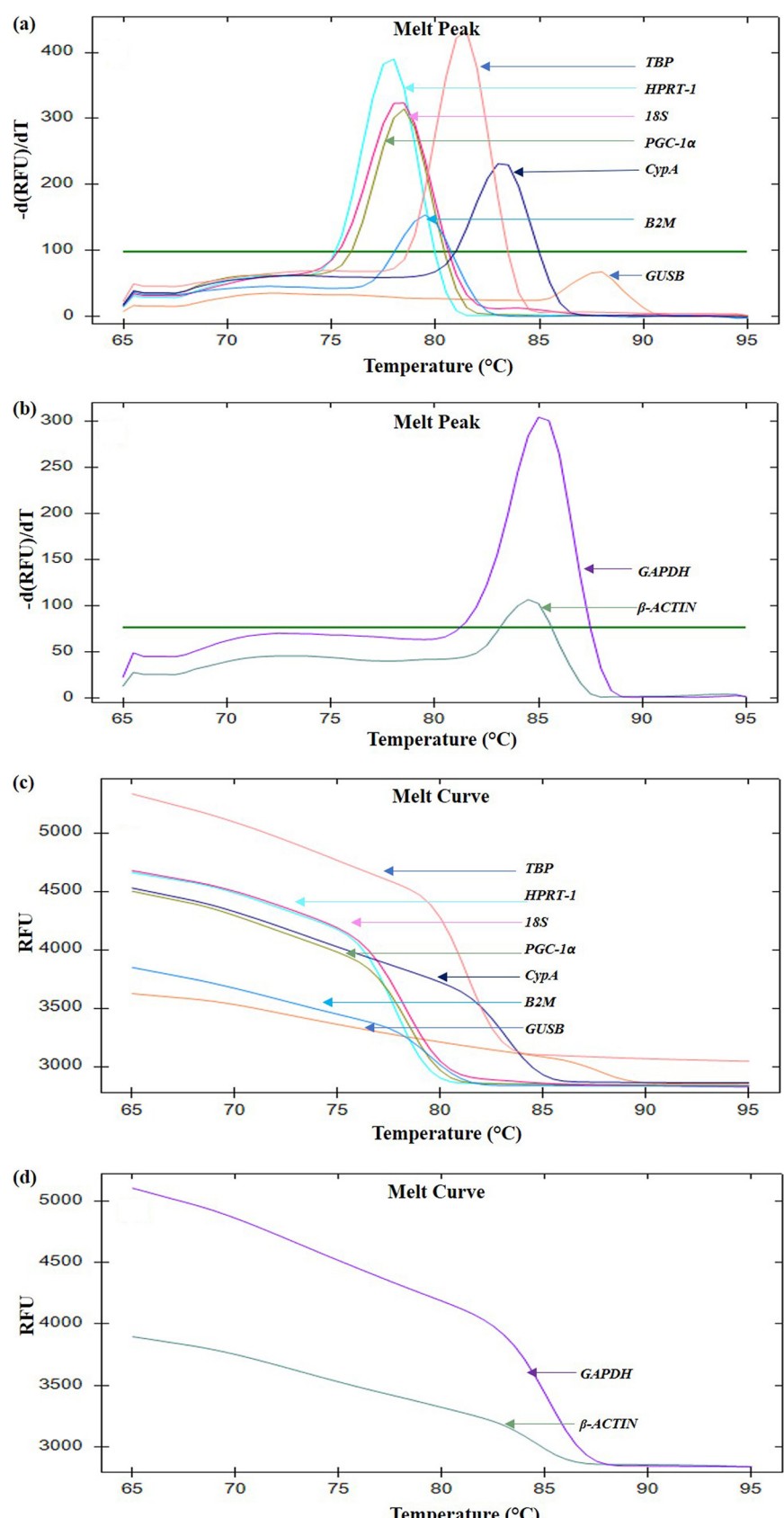

**FIG 1** Assessment of dissociation characteristics of candidate reference genes using melt curve analysis. (a and b) Melt peak of *TBP*, *CypA*, *B2M*, 18S, *PGC-1α*, *GUSB*, and *HPRT-1* (a) and *GAPDH*

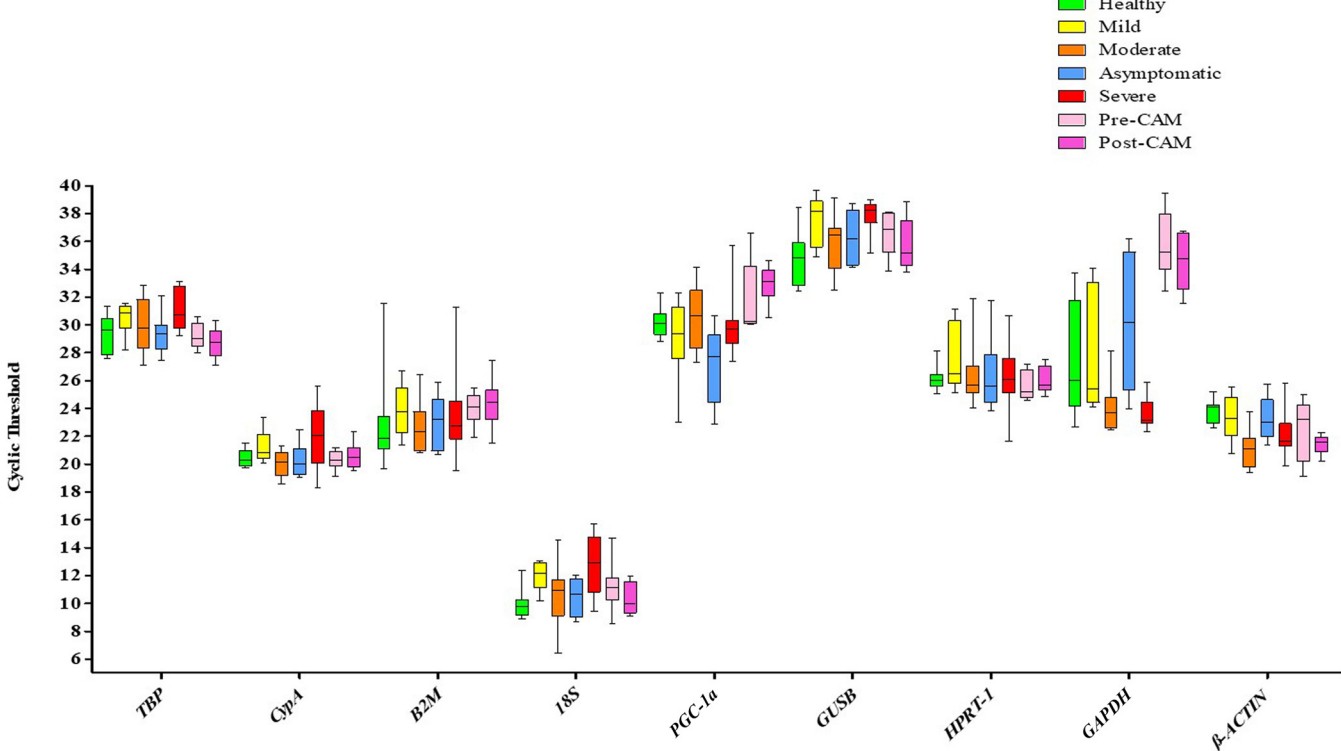

**FIG 2** Analysis of candidate reference genes reveals significant heterogeneity in gene expression among patients with various disease severities. Boxplots of mean $C_T$ values (y axis) for candidate reference genes (x axis) used in this study are shown. The line across the box represents the median $C_T$ value, whereas whiskers indicate the range of $C_T$ values obtained in a particular group of patients. Data represent a total of 56 subjects (n = 56; male, 39; female, 17; 8 subjects/group) including the healthy individuals.

belonging to seven clinically defined groups (healthy, asymptomatic, mild, moderate, severe, pre-CAM, and post-CAM). All nine candidate reference genes were within the dispersal range of threshold cycle ($C_T$) value (range, >6.43 to 39.69) and also fold increase in PCR product per cycle, i.e., the amplification efficiency was within the accepted range (90 to 100%).

Based on the severity of SARS-CoV-2 or associated fungal infection, we noticed significant differences in the expression of candidate reference genes, between and within groups across the disease spectrum (Fig. 2). Surprisingly, the most commonly used internal control gene in COVID-19 studies, i.e., *GAPDH*, showed significant variations within healthy subjects and also between healthy and COVID-19 subjects who had a moderate or severe infection or CAM. For *β-ACTIN*, even though expression levels were comparable in healthy, mild, and asymptomatic individuals, substantial variations in individuals with moderate or severe infections or CAM were observed. Likewise, another commonly used housekeeping gene, peroxisome proliferator-activated receptor gamma (PPARG) coactivator 1 alpha (*PGC-1α*), also demonstrated significant variations among asymptomatic infections or CAM.

For 18S and glucuronidase beta (*GUSB*), the ranges of $C_T$ values were comparable across groups. However, these genes consistently were expressed either very early (18S, 6.43 to 15.73; 95% confidence interval [CI], 10.56 to 11.53) or very late (*GUSB*, 32.40 to 39.69; 95% CI, 35.88 to 36.94). Ideally, for an internal reference gene, a moderate expression level as exemplified by a $C_T$ value of 15.0 to 30.0 is desirable to accurately represent the quantitative expression of the genes of interest (26). As shown in Fig. 2, only TATA-box binding protein

**FIG 1** Legend (Continued)

and *β-ACTIN* (b). (c and d) Melt curve of *TBP*, *CypA*, *B2M*, 18S, *PGC-1α*, *GUSB*, and *HPRT-1* (c) and *GAPDH* and *β-ACTIN* (d). A single melt peak and a single melt curve for each individual gene indicate the generation of a single amplicon following primer amplification. RFU, relative fluorescence unit.

(*TBP*), *CypA*, $\beta$-2-microglobulin (*B2M*), and *HPRT-1* have comparable mean $C_T$ values across the SARS-CoV-2 spectrum and CAM. Nonetheless, the identity of the most suitable reference gene is not immediately obvious.

**Application of statistical algorithms identifies *CypA* and *TBP* as the most appropriate reference genes for SARS-CoV-2 infection and associated mucormycosis.** Considering the significant variations in $C_T$ values observed among patients (inter- or intragroup), we next employed popular statistical algorithms such as the comparative delta-$C_T$ method, BestKeeper, GeNorm, NormFinder, and RefFinder for objectively determining the stability of reference genes. These algorithms capture various aspects of expression data including gene stability, inter- and/or intragroup variations, standard deviations (SDs), etc., to indicate a suitable reference gene.

Using the comparative delta-$C_T$ method, we observed that *GAPDH* and *PGC-1α* showed the highest standard deviations across COVID-19 and CAM cases as evident from the cumulative standard deviation data for individual genes (Fig. 3a). *CypA* has the lowest average and the lowest cumulative standard deviation in expression, followed by 18S and *TBP*. Another tool, BestKeeper, on the other hand indicated *CypA* and *TBP* as the two most stable reference genes. We also used popular tools, namely, NormFinder and GeNorm, both of which revealed *GAPDH* to be the least suitable gene based on the highest arithmetic mean of all pairwise variations (indicated by M value in GeNorm) and the highest stability value, which is inversely related to the stability of the reference gene (i.e., the most stable gene has the lowest stability value). Both of these tools identified *CypA* as the most suitable reference gene followed by 18S. Remarkably, neither of the two most popularly used internal control genes in COVID-19, i.e., *GAPDH* and $\beta$-*ACTIN*, showed expression consistency across the patients with various severities of disease. All the above tools identified *CypA* as the ideal reference gene across the disease spectrum by consensus (Fig. 3).

We next applied a widely used algorithm, RefFinder (27), that considers the output of four other algorithms (comparative delta-$C_T$ method, BestKeeper, GeNorm, and NormFinder) as input data to yield a final comprehensive ranking based on the geometric mean (GM) of ranking values. Based on RefFinder analysis, *CypA* emerged as the most suitable gene with the lowest cumulative GM (10.35) (GM has an inverse relation with gene stability; the lower the GM, the higher the stability). This value is less than half of those of the second and third most suitable reference genes identified here, i.e., 18S (GM = 22.92) and *TBP* (GM = 26.43), respectively. *GAPDH* has a cumulative GM (53.51) over five times higher than that of *CypA* and therefore emerged as the least suitable reference gene. $\beta$-*ACTIN*, the other widely used reference gene in COVID-19 and other viral infections, could be ranked only 5th out of the nine candidate genes evaluated (Table 1).

**Gene expression analysis of candidate genes *NRF2*, *IL-6*, and *IL-15* based on normalization with *CypA* and *GAPDH* confirms the suitability of *CypA* as an internal control.** We next evaluated the relevance of having an appropriate internal control by comparing differences between gene expression of select candidate genes based on normalization with *CypA* and *GAPDH*, also previously used in COVID-19 (28, 29). We selected key genes whose gene expression profile have been previously known from RNA sequencing (30–33). This guided us in comparing the qRT-PCR-based gene expression data obtained following normalization with internal control genes. Since no molecular studies were available for CAM patients, we focused on studies featuring COVID-19 severity and selected nuclear factor erythroid 2-related factor 2 (*NRF2*), interleukin-6 (*IL-6*), and *IL-15* as the candidate genes for this analysis (30–33).

*NRF2* is a master regulator for an array of antioxidant responses, and excessive oxidative stress is a well-documented hallmark of COVID-19 pathophysiology (34). However, in SARS-CoV-2 infection, *NRF2* expression is dysregulated and not sufficiently induced to mount an effective antioxidant response (35). The relative suppression of *NRF2* expression is a viral strategy to hijack host protective responses (35), and pharmacological induction of *NRF2* expression is proposed to be an effective strategy against COVID-19 (36).

We evaluated how *NRF2* expression would vary in patients based on the choice of the

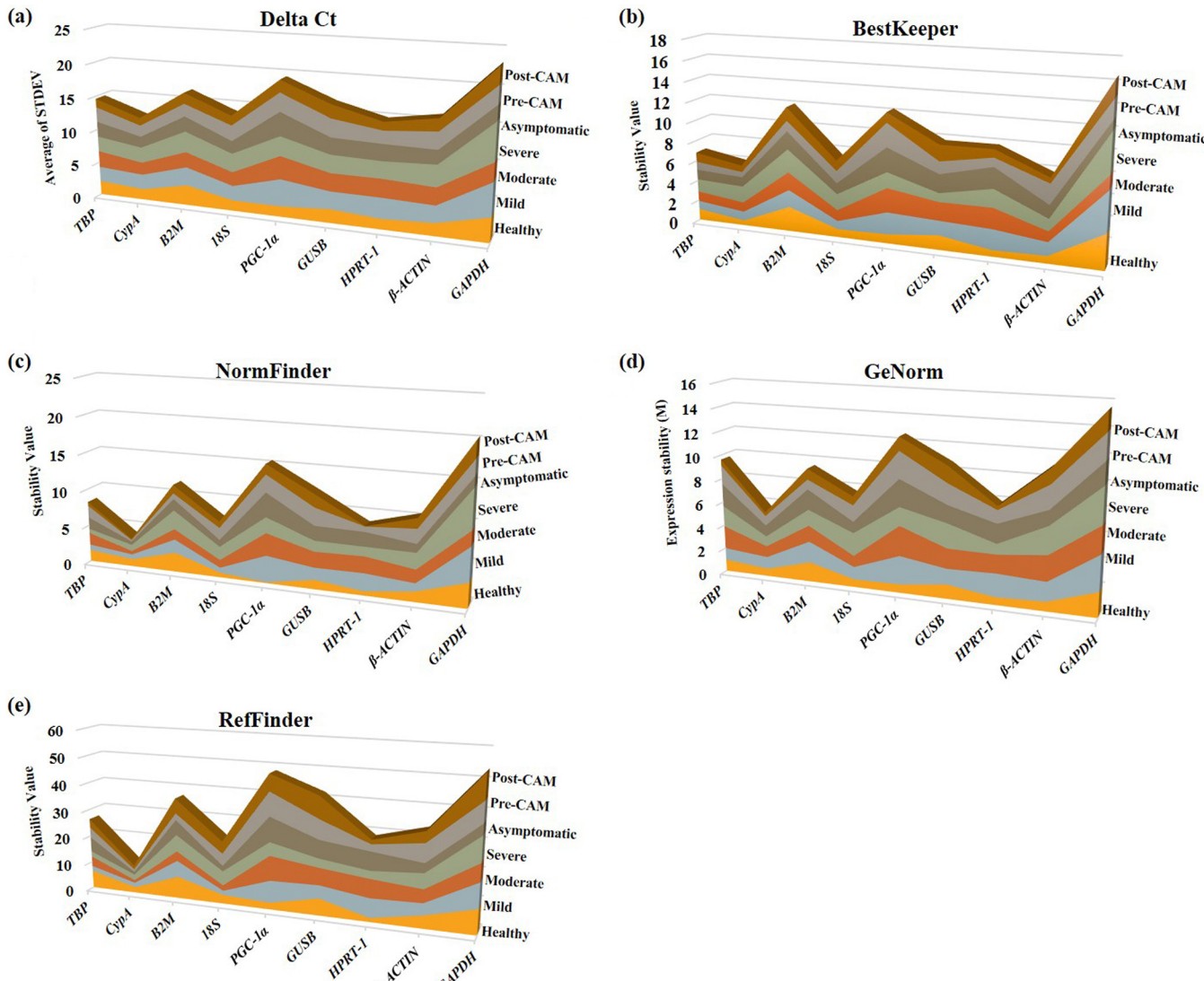

**FIG 3** Comparison of candidate reference genes based on cumulative ranking score using different statistical algorithms. (a) Intragroup average standard deviations for individual patient groups were obtained using the delta-$C_T$ method and are added to obtain a cumulative average standard deviation ($y$ axis). The highest cumulative average standard deviation (i.e., *GAPDH*) indicates the lowest gene stability ($x$ axis). (b) The graph represents a cumulative stability value ($y$ axis) obtained by adding intragroup stability value based on coefficient of correlation ($r$) and standard deviation using the BestKeeper algorithm. The candidate gene yielding the lowest cumulative value (i.e., *PGC-1α*) indicates the highest stability. (c) In the NormFinder algorithm, intragroup stability values ($y$ axis) based on coefficients of variation are added to obtain a cumulative stability value. The candidate gene yielding the lowest cumulative stability value (i.e., *CypA*) indicates the most stable reference gene. (d) In the GeNorm algorithm, intragroup expression stability (M value) based on the average pairwise variation of an individual gene with all other genes is used as a control to obtain a cumulative expression stability ($y$ axis). The gene yielding the highest M value (i.e., *GAPDH*) indicates the lowest stability. (e) In the RefFinder algorithm, intragroup geometric means ($y$ axis) based on geometric mean of ranking values calculated by all algorithms above are added to obtain a cumulative geometric mean of ranking values. *CypA* yielded the lowest cumulative geometric mean, indicating the highest stability, and therefore emerged as the most suitable reference gene.

internal control. In our patient cohort, *NRF2* expression based on *CypA* normalization is consistent with the previous reports wherein *NRF2* expression is low and relatively suppressed (only 1.65- to 4.2-fold change compared to healthy subjects even in the high-oxidative-stress inflammatory environment [30]). Apparently, there was no statistically significant difference in *NRF2* expression across the spectrum of COVID-19 and mucormycosis. However, we observed ~12-fold induction only in the pre-CAM group (Fig. 4). Normalization with *GAPDH*, however, could not accurately capture the heterogeneity of the COVID-19 spectrum and that of CAM patients, and a significantly higher induction of *NRF2* (1.3- to 268-fold in asymptomatic, mild, and moderate groups) was observed in COVID-19 patients (Fig. 4). Moreover, in pre- and post-CAM groups, the *NRF2* induction based on *GAPDH* normalization was ~600-fold and ~230-fold, respectively (Fig. 4). In only the severely sick group, there was no

**TABLE 1** Comprehensive ranking based on geometric mean (GM) obtained using RefFinder analysis[a]

| Gene | GM for group: | | | | | | | Cumulative geometric mean | Comprehensive ranking |
|------|---------|------|----------|--------|--------------|---------|----------|---------|---------|
| | Healthy | Mild | Moderate | Severe | Asymptomatic | Pre-CAM | Post-CAM | | |
| *CypA* | 2.00 | 1.73 | 1.00 | 2.21 | 1.00 | 1.00 | 1.41 | 10.35 | 1 |
| 18S | 3.22 | 1.57 | 2.00 | 5.23 | 2.00 | 4.47 | 4.43 | 22.92 | 2 |
| *TBP* | 6.48 | 1.86 | 3.72 | 1.97 | 5.42 | 3.76 | 3.22 | 26.43 | 3 |
| *HPRT-1* | 1.41 | 7.00 | 6.73 | 2.82 | 6.65 | 2.45 | 1.57 | 28.63 | 4 |
| *β-ACTIN* | 4.61 | 4.23 | 4.79 | 5.47 | 3.41 | 6.70 | 3.98 | 33.19 | 5 |
| *B2M* | 8.00 | 6.00 | 3.57 | 6.05 | 5.48 | 2.45 | 5.44 | 36.99 | 6 |
| *GUSB* | 6.19 | 4.73 | 6.19 | 3.25 | 6.51 | 7.14 | 8.00 | 42.01 | 7 |
| *PGC-1α* | 2.51 | 8.00 | 9.00 | 4.90 | 9.00 | 8.74 | 6.19 | 48.34 | 8 |
| *GAPDH* | 9.00 | 9.00 | 6.40 | 9.00 | 3.87 | 7.24 | 9.00 | 53.51 | 9 |

[a]The lowest cumulative geometric mean corresponds with the highest comprehensive ranking or suitability.

apparent difference in *NRF2* expression based on normalization with *CypA* or *GAPDH*. Clearly, the normalization of *NRF2* with *GAPDH* would be rather misleading and would not reflect the possibility of *NRF2* activation-based therapies proposed by several independent studies (17, 18, 30).

To further determine the suitability of *CypA* as an ideal reference gene, we next studied *IL-6*, which is transcriptionally inhibited by *NRF2* to block the proinflammatory responses (37, 38). A negative relationship therefore exists between *NRF2* and *IL-6*, and a high *NRF2* expression would lead to suppression of *IL-6* expression. Nonetheless, the normalization with *GAPDH* as a reference gene produced a high-*NRF2*–high-*IL-6* expression phenotype as both *IL-6* and *NRF2* showed massive induction (>400-fold) in patients with mild infection and in pre- and post-CAM patients. Moreover, *GAPDH* normalization did not reveal any significant differences in the expression of *IL-6* between asymptomatic and severely infected COVID-19 patients (Fig. 5) despite the fact that inflammation is minimal in asymptomatic patients and very high in patients having severe COVID-19 (39).

On the other hand, application of *CypA* as a reference gene indicated an overall higher *IL-6* expression and low *NRF2* expression in all groups. The highest *IL-6* expression was observed in severe COVID-19 patients who had the lowest *NRF2* expression. Furthermore, expression of *IL-6* in mild, moderate, and severe cases differed significantly, as inflamma-

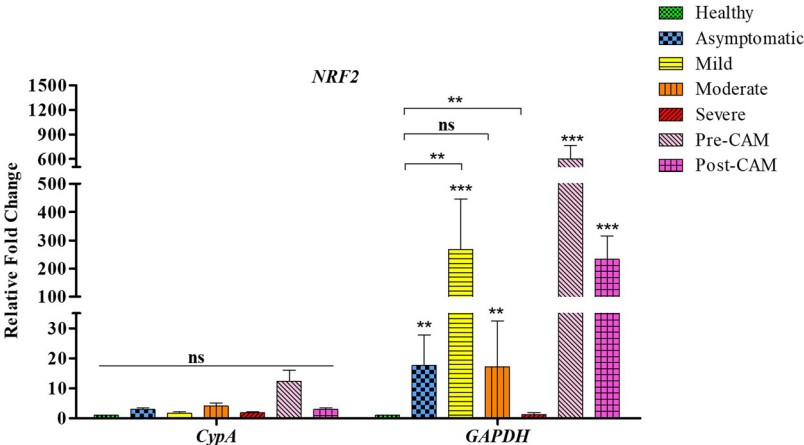

**FIG 4** Gene expression profile of *NRF2* based on use of *CypA* or *GAPDH* as an internal control. *NRF2* expression based on *CypA* normalization reveals that no significant difference exists in its expression across various patient groups. This is consistent with the RNA sequencing-based expression wherein *NRF2* expression is relatively low even in the high-oxidative-stress environment reported for COVID-19 and CAM (30, 31). On the other hand, normalization with *GAPDH* in our study reveals significant expression heterogeneity that varies from 2-fold to over 600-fold across the spectrum of COVID-19 and CAM cases. This does not match with RNA sequencing-based data from published works (31) and erroneously creates the impression of *NRF2* overexpression. Healthy subjects represent the baseline *NRF2* expression. Data represent mean ± SEM for assays performed with at least eight subjects in duplicate. Significance is ascertained by using a one-way ANOVA with Kruskal-Wallis test with Dunn's *post hoc* corrections (n = 8/ group; *, P value < 0.05; **, P value < 0.01; ***, P value < 0.001; ns, nonsignificant).

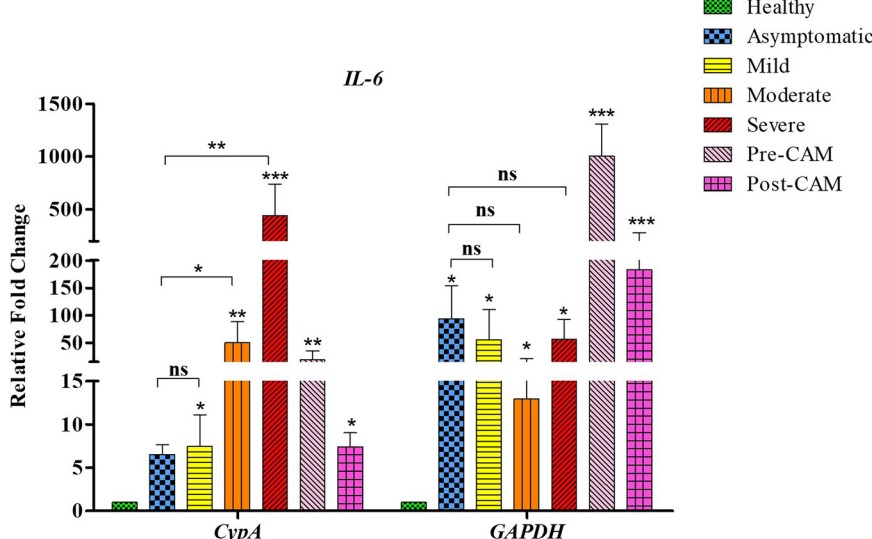

**FIG 5** Gene expression profile of *IL-6* based on normalization with *CypA* or *GAPDH* as an internal control gene. With *GAPDH*, we observed similar expression levels among asymptomatic, mild, moderate, and severe infections. On the other hand, RNA sequencing data indicate a severity-dependent increase in *IL-6* expression going from mild to severe infections (32–34). Only *IL-6* expression derived from normalization with *CypA*, therefore, is in agreement with previously published studies. The *y* axis represents differences in the fold change based on data normalization *vis-à-vis CypA* or *GAPDH* in various groups (*x* axis) of COVID-19 and CAM patients. Significance is ascertained with respect to healthy subjects by using a one-way ANOVA with Kruskal-Wallis test with Dunn's *post hoc* corrections (*, *P* value < 0.05; **, *P* value < 0.01; ***, *P* value < 0.001; ns, nonsignificant).

tion levels vary significantly among these groups, with asymptomatic patients having the lowest inflammation and the severely sick reportedly having the highest inflammation (39). Our observations based on *IL-6* expression derived from normalization with *CypA* and not *GAPDH* were in accordance with the plasma IL-6 levels that we determined to support the role of *CypA* as an ideal reference gene (Fig. 6a). With *GAPDH* normalization, even asymptomatic patients exhibited significant induction of *IL-6*, and the *IL-6* expression was also comparable with that of patients having moderate infections and those with CAM. To further delineate the relationship of gene expression with inflammation status, we next determined the plasma levels of C-reactive protein (CRP), which is an established clinical marker of systemic inflammation among COVID-19 patients. We observed a direct relationship between CRP and disease severity (40), with asymptomatic subjects having the lowest CRP levels and severe COVID-19 patients having the highest (Fig. 6b). The CRP and IL-6 levels correlated well with the gene expression derived from *CypA* normalization. In addition to *IL-6* and *NRF2*, we also scored for *IL-15* gene expression and obtained similar results wherein the normalization with *CypA*, and not *GAPDH* (Fig. 7), corresponds fully with the RNA sequencing data for *IL-15* (31). Thus, in summary, only data normalized with *CypA* yielded a gene expression pattern that broadly concurs with the clinical status, substantiating its utility as an ideal reference gene in COVID-19 and CAM patients.

## DISCUSSION

The availability of qRT-PCR instrumentation in clinics/molecular biology laboratories in even low-income countries affords unique opportunities to study gene expression that can recognize and differentiate discrete gene subsets to predict disease outcome or response to therapy. Unfortunately, there is no universal reference gene for qRT-PCR studies, necessitating the evaluation of reference genes for their suitability for an individual condition. The premise of our study is to select an appropriate reference gene that could capture broad-spectrum health manifestations of COVID-19 in patients (asymptomatic, symptomatic, mild, moderate, and severe), including coinfections with mucormycosis, with the final goal of benefiting patient-centric translational research.

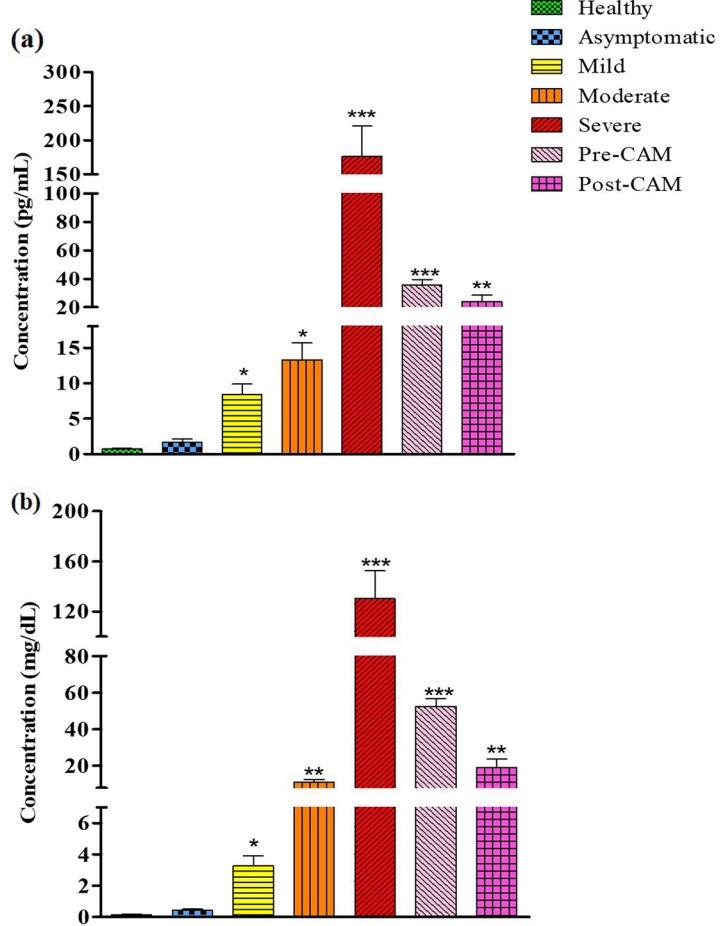

**FIG 6** IL-6 and C-reactive protein (CRP) levels in plasma across the spectrum of COVID-19 severity and CAM. Asymptomatic patients have the lowest and patients having severe COVID-19 have the highest IL-6 (a) and CRP (b) levels (*x* axis). These data are consistent with the gene expression profile derived from normalization with *CypA*. Significance is ascertained by using a one-way ANOVA with Kruskal-Wallis test with Dunn's *post hoc* corrections (*n* = 68; *, *P* value < 0.05; **, *P* value < 0.01; ***, *P* value < 0.001).

Both COVID-19 and CAM are recently emerged infections and warrant investigations of molecular mechanisms to develop better management and treatment strategies. Nonetheless, a formal determination of reference genes for qRT-PCR analysis has not yet been investigated.

An ideal reference gene requires that its gene expression should be minimally regulated across the disease spectrum with minimal influence of patient heterogeneity in terms of age/sex or severity, treatment, etc. For this purpose, we evaluated nine housekeeping genes commonly used for normalization of gene expression and determined their expression across patient groups with a broader age profile (age range, male, 17 to 75 years; female, 16 to 70 years; Table 2) in COVID-19 and CAM cases. The candidate reference genes were selected because these genes are constitutively expressed in all cells and perform different functions, thus minimizing the overrepresentation of genes belonging to the same biological pathway. Indeed, a large number of these nine candidate reference genes displayed variations in gene expression based on patient heterogeneity and therefore did not satisfy the prerequisites of a suitable reference gene as substantiated by various algorithms used in our study. Given that a suitable reference gene should display stable expression, only *CypA* demonstrated consistent expression across all patient groups and accordingly emerged as the most suitable reference gene in this study, followed by *TBP*.

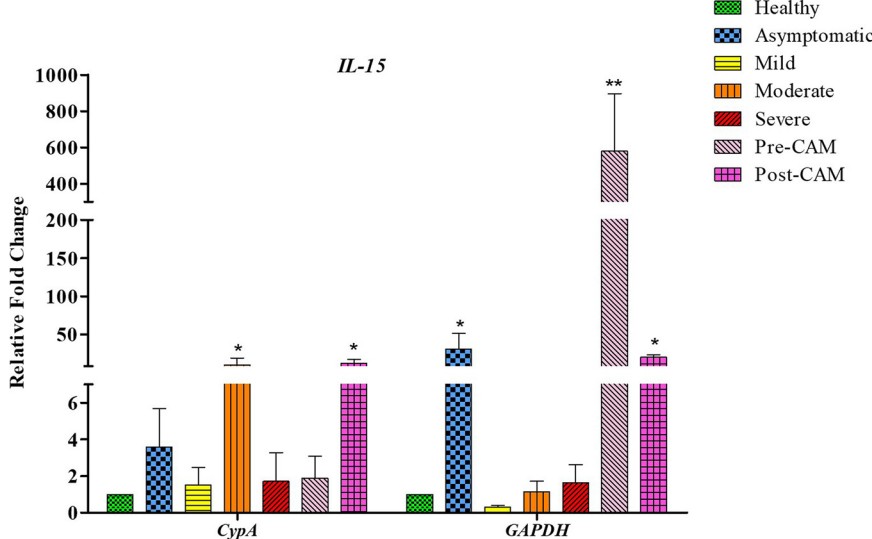

**FIG 7** Gene expression profile of *IL-15* gene based on the use of *GAPDH* or *CypA* as the internal control gene. Normalization with *GAPDH* does not concur with RNA sequencing-based data which show that *IL-15* expression increases significantly from mild to moderate disease and then returns to baseline expression in the severely sick (31). In the case of normalization with *CypA*, the expression pattern concurs with RNA sequencing data by demonstrating an increase from mild to moderate groups and a reduction in the severely sick. The *y* axis represents differences in the fold change based on data normalization in various groups (*x* axis) of COVID-19 and CAM patients. Significance is ascertained by using a one-way ANOVA with Kruskal-Wallis test with Dunn's *post hoc* corrections (*, *P* value < 0.05; **, *P* value < 0.01; ***, *P* value < 0.001).

First, all the patients in our study had a clinically confirmed diagnosis of mucormycosis or COVID-19. The patients' classification into various categories (mild to severe) was based on well-established international guidelines on clinical severity that are independent of viral load (41). To isolate RNA, we used PBMCs as they are one of the most accessible, commonly used, and easy-to-use cell types, especially in clinical settings. Further, the gene expression profile in PBMCs correlates well with the systemic gene expression changes in the disease condition (42). These features make them an attractive candidate to investigate differential gene expression, particularly for determination of quantitative traits involved in phenotypic outcomes (43).

To minimize the variations in gene expression, housekeeping genes are usually considered reference genes. But even housekeeping genes are involved in a plethora of cellular pathways, which may become dysregulated in patients based on a variety of factors, including severity (44–46). Therefore, heterogeneity in gene expression among patients is natural, expected, and commonly present, thereby necessitating the deter-

**TABLE 2** Description of subjects in study cohort used for gene expression analysis[a]

| Characteristic | Value by gender: | |
| --- | --- | --- |
| | Male (*n* = 39) | Female (*n* = 17) |
| Age (range in yr) | 17–75 | 16–70 |
| Median age (yr) | 45 | 35.5 |
| | | |
| No. of individuals with comorbidity | | |
| Diabetes | 16 | 3 |
| Hypertension | 2 | 1 |
| Anemia | 1 | 0 |
| Hyperthyroidism | 1 | 1 |
| Malignancy | 1 | 0 |
| Heart disease | 1 | 0 |

[a]Patient cohort of 56 individuals (39 males and 17 females) enrolled from different groups (healthy, asymptomatic, mild, moderate, severe, pre-CAM, and post-CAM) along with their comorbidity status. Overall median age was 42 years.

mination of the most suitable reference gene. *GAPDH* and *β-ACTIN* are among the most widely used reference genes for qRT-PCR studies, including those of COVID-19. For CAM patients, however, there is not a single gene expression or RNA sequencing-based study performed to date. In our study, *GAPDH* emerged as the least suitable reference gene as significant variations in its expression are observed in patients with various severities of disease. This could be because *GAPDH*, in addition to glycolysis, is also involved in diverse cellular functions including regulation of inflammation (45). Indeed, we observed a differential inflammation status among COVID-19 patients as also exemplified by significant differences in the levels of IL-6 and CRP (46, 47) (Fig. 6a and b). *β-ACTIN*, another regularly used reference gene, including in COVID-19, was also found unsuitable and was ranked 5th out of the nine candidate genes. The other candidate reference genes had very high expression levels, such as 18S, or very low expression levels, such as *GUSB*, in both healthy and diseased subjects, making them unsuitable.

Based on our results, *CypA* has the most stable expression across the spectrum of COVID-19 cases. Consequently, *CypA* emerged as the most appropriate reference gene in qRT-PCR-based analysis of COVID-19 and mucormycosis. 18S emerged as the second most stable gene, but it is expressed very early, while a medium- to high-level expression is desirable for an ideal reference gene (26), and hence, this limits the utility of 18S as a reference gene in qRT-PCR. On the other hand, *TBP* consistently showed a moderate expression across the COVID-19 spectrum and has only a marginal difference between its cumulative geometric means (26.43) *vis-à-vis* 18S (22.92), making it a better choice than 18S. As per the standard guidelines for reference gene selection for qRT-PCR, $C_T$ values of ≤40 are acceptable for determination of quantitative PCR (qPCR) efficiency and intra- and inter-assay variability (48). Further, most qRT-PCR-based COVID-19 diagnostics kits approved by WHO, the CDC, and other national and international agencies consider a $C_T$ of ≤40 as the cutoff value for effective diagnosis. In our study, irrespective of differences in clinical severity among subjects, the overall $C_T$ values were within the acceptable range for each group (dispersal range, 6.43 to 39.69). These variations in $C_T$ values among and within groups of patients reflect a natural heterogeneity regularly observed in the clinic.

The mean 260/280 ratio for RNA in our experiments ranged from 1.94 to 1.98 for healthy, asymptomatic, and mildly infected subjects and 1.73 to 1.79 in the case of CAM and severe COVID-19 patients. There were no statistically significant differences in the mean 260/280 ratio across the groups (see Fig. S2 in the supplemental material). While a 260/280 ratio of ~2.0 is desirable, 260/280 ratio values of ~1.6 to 1.7 have been often used for downstream applications based on the complexity of the samples (49–54). The minor and statistically nonsignificant variations in 260/280 ratio among various groups in our study did not affect the conclusions as determined by reanalysis of data ($n$ = 49 patients), excluding samples with a 260/280 ratio of <1.7 ($n$ = 7 patients) (data not shown).

Selection of unsuitable reference genes may weaken the detection sensitivity of the target genes and even result in inaccurate outcomes (55). To determine how gene expression varies based on normalization with *CypA* and *GAPDH*, we looked at genes such as *NRF2*, *IL-6*, and *IL-15*, whose expression profiles based on RNA sequencing were previously reported in COVID-19 patients (30–33). This was because several studies have shown that RNA sequencing results correlate well with the qRT-PCR data (56, 57). Genes were selected based on their role in redox and inflammation as redox dysregulation and oxidative stress are the hallmarks of various infections including COVID-19 (1). *NRF2* is a major redox regulator that is significantly induced in response to oxidative stress. Nonetheless, as a component of viral strategy to hijack host protective responses, *NRF2* expression is dysregulated during COVID-19, is not sufficiently upregulated, and remains relatively suppressed/minimally induced to mount an effective antioxidant response (30, 35). Based on normalization with *CypA*, we obtained a relatively stable *NRF2* induction profile in mild (<2-fold), asymptomatic (<3-fold), moderate (<5-fold), pre-CAM (<13-fold), and post-CAM (<3-fold)

groups. Also, there was no significant difference in the *NRF2* expression across various groups based on normalization (Fig. 4). The slightly higher *NRF2* levels observed in the CAM group may be a response to mucormycosis, and *NRF2* expression returns closer to baseline following reduction in fungal burden postsurgery (post-CAM). On the other hand, normalization with *GAPDH* highlighted significant expression heterogeneity (2-fold to over 600-fold) across the spectrum of COVID-19 and CAM. Significant upregulation of *NRF2* was observed in mild (>260-fold), asymptomatic (>17-fold), moderate (>17-fold), pre-CAM (>600-fold), and post-CAM (>230-fold) groups. This is in contrast to the RNA sequencing studies, showing relatively suppressed expression of *NRF2* in COVID-19 and with stimulation of *NRF2* expression proposed as a therapeutic strategy (30, 58) Evidently, normalization with *GAPDH* erroneously creates the impression of *NRF2* overexpression, especially in mild infections and CAM. In the severe group only, we obtained similar results from normalization with *GAPDH* and *CypA* (<2-fold induction in both cases).

The status of *CypA* as an ideal reference gene was further confirmed with the help of additional genes *IL-15* and *IL-6*. *NRF2* and *IL-6* have a negative relationship as *IL-6* expression is transcriptionally inhibited by *NRF2* (37, 38). Since *NRF2* expression levels are low in the COVID-19 spectrum, high *IL-6* expression is expected in these groups. Only when we used *CypA* as a reference gene, we obtained an overall higher *IL-6* and low *NRF2* expression phenotype. That upon normalization with *GAPDH* both *IL-6* and *NRF2* showed massive induction (>400-fold) in the case of patients with mild infection and in pre- and post-CAM patients reinforces the idea that *GAPDH* is not a suitable reference gene for the COVID-19 spectrum. Analysis of IL-6 levels concurs well with the *CypA*-normalized *IL-6* expression, and those levels were also consistent with the CRP levels, which further supports our findings and the suitability of *CypA* as an ideal reference gene.

Therefore, it is clear that *CypA* is the most suitable reference gene to use for COVID-19 and CAM and that only normalization with *CypA* can accurately capture the expression heterogeneity across the disease spectrum. Use of *CypA* would yield an accurate gene expression profile to guide development of further therapeutic and prognostic modalities in COVID-19 and CAM. To the best of our knowledge, the current investigation is the only systematic analysis of the stability of housekeeping genes during the course of COVID-19 and mucormycosis. The results clearly establish that expression of a housekeeping gene may vary due to infection and reinforce the idea of careful selection of housekeeping genes post-validation.

## MATERIALS AND METHODS

**Study subjects.** Subjects of different ages (25 to 60 years) and genders having SARS-CoV-2 infection (mild, moderate, severe, and asymptomatic) and COVID-19-associated mucormycosis, i.e., presurgery (pre-CAM) and postsurgery (post-CAM), along with healthy controls, were considered (total subjects = 56, 8 subjects/group; Table 2) The COVID-19 status of the subjects was confirmed by the presence of viral content determined by using Xpert Xpress SARS-CoV-2 (Cepheid, USA) or the TrueNat system (Molbio Diagnostics, India) approved by the Indian Council of Medical Research (ICMR) for COVID-19 diagnostics. Briefly, the patients were evaluated for the presence of *N2* and *E* genes of SARS-CoV-2 (Xpert Xpress SARS-CoV-2). In the case of TrueNat, initial screening was based on the presence of the *E* gene, followed by that of the *RdRp* gene in the case of positive screening results. Mucormycosis was confirmed by performing tissue biopsy and fungal staining (KOH mount). The presence of fungal hyphae on the KOH mount indicated the presence of mucormycosis and was confirmed in the hospital by a clinical mycologist. We collected blood samples at the time of admission of patients by taking informed consent. Individuals in the healthy category were COVID-19 negative as confirmed by qRT-PCR, with no history of COVID-19 and comorbidities (diabetes, influenza, cough, tuberculosis [TB], etc.).

**Criteria of disease and clinical severity.** All the cases of COVID-19 and CAM were clinically confirmed, and severity of COVID-19 infection was ascertained per the institutional clinical definitions that follow the same guidelines as recommended by the Indian Council of Medical Research (ICMR) and the Government of India. These guidelines are universal in nature and are consistent with the guidelines of other global bodies such as the U.S. NIH and WHO. The broad categorization of patients is indicated below:

 (a) Mildly symptomatic: proven SARS-CoV-2 infection with mild symptoms, i.e., fever and cough without shortness of breath or hypoxia.

(b)  Moderately sick: proven SARS-CoV-2 infection with moderate disease, i.e., fever with cough and/or shortness of breath with respiratory rate of $\geq$24/minute or oxygen saturation (SpO$_2$) of 90% to $\leq$93% on room air.

(c)  Severe: proven SARS-CoV-2 infection with moderate to severe disease, i.e., respiratory failure requiring mechanical ventilation and clinical evaluation by the attending physician. Respiratory rate of >30/min, breathlessness, or SpO$_2$ of <90% on room air.

(d)  Asymptomatic: proven SARS-CoV-2 infection but without any apparent symptoms.

(e)  Pre-CAM: confirmed mucormycosis in patients having COVID-19 or immediately after recovery from COVID-19 ascertained by RT-PCR report.

(f)  Post-CAM: patients having CAM and reporting to the hospital at an advanced stage of infection or who were referred cases who had to undergo surgery immediately and were still under treatment for CAM at the time of sample collection.

The design of the study did not influence the clinical care or treatment for any of the patients.

**Isolation of PBMCs, RNA isolation, and cDNA synthesis.** Peripheral blood mononuclear cells (PBMCs) were isolated using density gradient centrifugation (Ficoll; HiMedia, India) as described earlier (59). Buffy coat containing PBMCs was removed carefully and washed twice with phosphate-buffered saline (PBS) at 1,500 rpm for 10 min. Approximately 10$^7$ cells were transferred to 1 mL TRIzol (Invitrogen, USA) for RNA isolation followed by downstream processing including qRT-PCR. RNA concentration (nanograms per milliliter) and purity ($A_{260}$/$A_{280}$) were determined spectroscopically (see Table S1 in the supplemental material) (Cytation; BioTek, USA). The mean 260/280 ratios for all patient groups were in the range of 1.7 to 2.0 (Fig. S2). cDNA synthesis was carried out using the iScript cDNA synthesis kit per the manufacturer's instructions (Bio-Rad, USA).

**Selection of candidate genes, primer design, and optimization of qRT-PCR conditions.** From literature review, we selected the nine most frequently used housekeeping genes, namely, *TBP*, *CypA*, *B2M*, 18S, *PGC-1α*, *GUSB*, *HPRT-1*, *β-ACTIN*, and *GAPDH*, to determine their suitability to act as reference genes (Table S2). Nuclear factor erythroid 2-related factor 2 (*NRF2*), interleukin-6 (*IL-6*), and interleukin-15 (*IL-15*) were used as the candidate genes to determine the suitability of the most and the least suitable reference gene identified in our study. Primer 3 and NCBI-BLAST (https://blast.ncbi.nlm.nih.gov/Blast.cgi) were used for extraction of gene sequences and primer design. For our study, we designed primers (18 to 24 bp, predicted melting temperature [$T_m$] = 50°C to 60°C, and GC content of 40 to 60%). We determined the amplification efficiency of individual primer pairs by performing gradient PCR. Briefly, a 20-$\mu$L volume of reaction mixture contained 10$\times$ PCR buffer, 500 $\mu$M deoxynucleoside triphosphate (dNTP), primer pair (500 nM each), 20 ng of the template (cDNA), and 1.5 U of *Taq* DNA polymerase (Bangalore GeNei, India). A gradient PCR was used with the following parameters: initial denaturation at 95°C for 3 min and 35 cycles of denaturation at 95°C for 30 s, annealing for 30 s at 58°C, 59°C, 60°C, and 61°C, and subsequent extension at 72°C for 30 s, followed by a final extension step for 5 min at 72°C. The amplified products were subjected to 1.2% agarose gel electrophoresis, and the temperature yielding the highest band intensity on the gel represented the best annealing temperature for an optimum amplification.

We used the SYBR green-based assay to quantify gene amplification by using a CFX Opus 96 qRT-PCR machine (Bio-Rad, USA). All the reactions were performed at least in duplicates with the final volume/reaction mixture of 20 $\mu$L having the following components: 10 $\mu$L of iTaq universal SYBR green supermix (2$\times$), 2 $\mu$L of the 1:10-diluted cDNA template with a final concentration of 10 ng/reaction, 0.5 $\mu$L each of forward and reverse primer (10 $\mu$M), and nuclease-free water to make the final volume. The cycle was programmed for an initial denaturation at 95°C for 5 min, followed by 39 cycles of denaturation at 95°C for 10 s, annealing at 60°C for 10 s, and extension at 72°C for 30 s. The melt curve was obtained by heating the amplicon from 65 to 95°C in a continuous acquisition mode and was analyzed to ascertain the specificity of the primers.

**Determination of CRP and IL-6 levels in patients.** C-reactive protein (CRP) and IL-6 levels were determined from plasma per the approved clinical protocols on hospital patients ($n$ = 8 to 12 subjects/group; total subjects = 68). Briefly, CRP in plasma samples was determined by particle-enhanced immunoturbidimetric assay where human CRP agglutinates with latex particles coated with monoclonal anti-CRP antibodies, using the Roche Cobas c-702 analyzer (Roche, Switzerland). The aggregates were determined turbidimetrically, and the analyzer automatically calculated CRP concentration in the sample (milligrams per deciliter) without manual intervention. Likewise, for IL-6 quantification, we used an automatic analyzer (Beckman Coulter, USA) that performs a one-step immunoenzymatic ("sandwich") assay. Briefly, the plasma sample is placed in a reaction vessel and incubated with paramagnetic particles coated with mouse monoclonal anti-human IL-6, blocking reagent, and the alkaline phosphatase conjugate. After incubation, materials are washed, followed by addition of chemiluminescent substrate. The light generated by the reaction is measured with an inbuilt luminometer. The concentration (picograms per milliliter) of analyte in the sample is determined from a multipoint calibration curve.

**Data analysis and algorithms for the selection of the most suitable reference gene.** Statistical analysis was performed using modules available in GraphPad ver. 5.01. The significance was determined using one-way analysis of variance (ANOVA) with Kruskal-Wallis analysis with Dunn's *post hoc* corrections (\*, $P$ value < 0.05; \*\*, $P$ value < 0.01; \*\*\*, $P$ value < 0.001; ns, nonsignificant). Data represented mean $\pm$ standard error of the mean (SEM) or standard deviation (SD). Different tools, namely, delta-$C_T$, BestKeeper, NormFinder, GeNorm, and RefFinder, were used to calculate the stability of each reference gene. All these software programs have inbuilt statistical tools to score for variations and decide on the most preferred genes.

**Ethics approval.** This study was approved by the institutional human ethics committee of the All India Institute of Medical Sciences (AIIMS), New Delhi (IEC-435/02.07.2021, RP-34/2021, and IEC-419/02.07.2021).

**Data availability.** All data are available from the corresponding authors upon reasonable request.

## SUPPLEMENTAL MATERIAL

Supplemental material is available online only.
**SUPPLEMENTAL FILE 1**, PDF file, 0.4 MB.

## ACKNOWLEDGMENTS

This work in part is supported by the AIIMS Intramural COVID-19 grants (A-Covid-54 and AC-44) and mucormycosis grant (A-Covid-78) to V. Saini. The research in the Laboratory of Infection Biology and Translational Research is supported by funding to V. Saini through the HarGobind Khorana Innovative Young Biotechnologist Award (BT/11/IYBA/2018/01), DST-SERB (CRG/2018/004510), Life Science Research Board, DRDO, India (LSRB-375/SH&DD/2020), and Consortium for One-Health to address zoonotic and transboundary diseases in India (BT/PR39032/ADV/90/285/2020). N. Kushwaha is supported by a DHR-HRD woman scientist grant (no. 12013/30/2020-HR), and S. Negi thanks CSIR for the fellowship.

We thank the staff of the Laboratory of Infection Biology and Translational Research, namely, Krishan Pal, Shailendra Kumar, Nidhi, and Shalini Sharma, for providing assistance in laboratory work and with sample collection and processing. We are also thankful to all the subjects and study volunteers for their participation to this study. We thank Bio-Rad India for providing qRT-PCR instrumentation available to us during the pandemic.

Concept, design, and overall supervision: V. Saini; clinical sample processing: S. Kumar, N. Kushwaha, S. Negi, and K. Gautam; data acquisition: S. Kumar, A. Ahmad, N. Kushwaha, and N. Shokeen; analysis and interpretation of data: S. Kumar, A. Ahmad, N. Shokeen, and V. Saini; sample collection and clinical inputs: N. Shokeen, A. Singh, P. Tiwari, R. Garg, R. Agarwal, A. Mohan, A. Trikha, and A. Thakar; drafting of the manuscript: S. Kumar and V. Saini wrote and edited the manuscript, and all authors provided input/comments on the manuscript; obtaining funding: V. Saini.

We have no conflicts of interest to declare.

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
