## [Reviewer comments · Microbiology Spectrum]

Microbiology Spectrum

Selection of ideal reference genes for gene expression analysis in COVID-19, and Mucormycosis infection

Sunil Kumar, Ayaan Ahmad, Namrata Kushwaha, Niti Shokeen, Sheetal Negi, Kamini Gautam, Anup Singh, Pawan Tiwari, Rakesh Garg, Richa Agarwal, Anant Mohan, Anjan Trikha, Alok Thakar, and Vikram Saini

Corresponding Author(s): Vikram Saini, All India Institute of Medical Sciences, New Delhi

Review Timeline:

Submission Date:	May 9, 2022
Editorial Decision:	July 2, 2022
Revision Received:	September 27, 2022
Accepted:	October 17, 2022

Editor: Zsolt Toth

Reviewer(s): Disclosure of reviewer identity is with reference to reviewer comments included in decision letter(s). The following individuals involved in review of your submission have agreed to reveal their identity: Gopal Naik Nenavath (Reviewer #2); Weiqiang Chen (Reviewer #3)

Transaction Report:

DOI: <https://doi.org/10.1128/spectrum.01656-22>

July 2, 2022

Dr. Vikram Saini
All India Institute of Medical Sciences, New Delhi
Biotechnology
New Delhi
India

Re: Spectrum01656-22 (Selection of ideal reference genes for gene expression analysis in Covid19, and Mucormycosis infection)

Dear Dr. Vikram Saini:

Thank you for submitting your manuscript to Microbiology Spectrum. We have completed our review and you will find the comments from the reviewers below.

We would like to invite the resubmission of a significantly-revised version that takes into account the reviewers' comments. When submitting the revised version of your paper, please provide (1) point-by-point responses to the issues raised by the reviewers as file type "Response to Reviewers," not in your cover letter, and (2) a PDF file that indicates the changes from the original submission (by highlighting or underlining the changes) as file type "Marked Up Manuscript - For Review Only". Please use this link to submit your revised manuscript - we strongly recommend that you submit your paper within the next 60 days or reach out to me. Detailed instructions on submitting your revised paper are below.

Link Not Available

Sincerely,

Zsolt Toth

Journals Department
Reviewers comments:

Reviewer #1 (Comments for the Author):

Comments

The manuscript by Sunil et al., has carried out very important study during this ongoing COVID-19 pandemic. Although, this study provides strong rationale for identification of appropriate reference genes in the viral infections but has some serious problems in the experiments and result interpretations. Overall, the results are not sufficient and convincing to their claims. Here are my comments for the authors.

1. The authors should show the viral genome content in the subjects analyzed.
2. I think it is difficult to interpretate the results in this kind of experimental settings since the reference gene expression levels

can be affected by several factors not only due to SARS-CoV-2 infections. For example, the age and health conditions of the subjects in the groups can be different.

3. Authors did not mention the quantity of cDNA was used in the RT-PCR reactions.

4. It was mentioned that the groups were categorized by institutional definitions, but the categorization of the groups must be approved by certain international bodies otherwise, it is hard to apply the findings of this study in general.

5. In Fig.2, some of the ct values for the GUSB or PGC-1 α or GAPDH in pre-CAM or post-CAM are above ct 30.0, in some cases even above ct 34.0. I think the ct values above 30 is not reliable for calculations. Moreover, the PCR was performed at annealing temperature of 60 degrees for all the reference genes. However, the primers could have better amplification efficiency at other temperatures as well. Not sure, if authors have performed gradient PCR and selected the annealing temperature for the primers.

6. A 260/280 ratio of ~2.0 is generally accepted as "pure" for RNA. In Table 1, the RNA quality for the groups of Severe, pre-CAM and post-CAM is below 2.0 and varies a lot compared to other groups. This is very important factor in the whole experiment as the downstream steps depends on the purity of RNA used for cDNA preparation.

7. To support the RT-qPCR data, authors should have analyzed the expression of these reference genes by Western blotting from the PBMC's collected.

8. Based on normalization with CypA, NRF2 expression level did not correlate with the disease severity in the groups.

Therefore, authors should have analyzed for at least 4-5 genes that may have differential expression in SARS-CoV-2 infections to prove that CypA is the suitable reference gene compared to GAPDH.

Reviewer #2 (Comments for the Author):

The manuscript by Kumar et al., entitled "Selection of ideal reference genes for gene expression analysis in COVID-19, and Mucormycosis infection" identified among 9 different housekeeping genes, the CypA gene as the most suitable reference gene, which accurately captured the heterogeneity of infection and yielded a stable expression across different conditions of COVID-19 severity and Mucormycosis infection (Covid-associated mucormycosis - CAM). Briefly, the authors tested the expression stability of nine different housekeeping genes (including TBP, CypA, B2M, 18S, PGC-1 α , GUSB, HPRT-1, β -ACTIN and GAPDH) in PBMC isolated from patients with different COVID-19 severity and CAM. The analysis of housekeeping genes during SARS-CoV-2 spectrum disease and CAM infection showed significant expression variations in most of candidate reference genes intra and inter-group. Only TBP, CypA, B2M and HPRT-1 had a comparable mean Ct value across SARS-CoV-2 spectrum and CAM infection. Then, the authors employed statistical algorithms to determine the stability indexes of each reference gene. GHPDH and PGC-1 α had more deviations. Nonetheless, CypA is identifies as the most suitable reference gene in Covid-19, and CAM infections.

Strength

Studying the co-infection of SARS-CoV-2 and mucormycosis is relevant based on existing public health issue and the lack of studies published in the field. Also, the selection of accurate reference genes in different disease outcomes could advance the field.

Weaknesses

The observations postulated in the paper are novel and interesting. However, the discussion is not robust enough to make convincing conclusion. The authors should show more data from published studies related to their results.

Below are comments to strengthen the manuscript.

Major comments

1) Figure 1- Figure 3: Figure 2 shows difference between healthy and Covid19 subjects with moderate or severe or CAM infections. GAPDH showed variations inter-groups, in which expression was comparable in health, mild and asymptomatic patients, while moderate and asymptomatic as well as Pre/Pos CAM had other expression patterns. How could you explain this pattern of high GAPDH variations inter-group (Figure 2) whether in Figure 3b the data shows GAPDH as a reference gene more stable than CypA?

2) Figure 4: As the health subjects represent the baseline NRF2 expression, is there significant difference in NRF2 expression among samples from moderate, asymptomatic and/or pre/post-CAM diseases when CypA was used as internal

control gene? If so, how can the NRF2 expression variability among them using CypA as internal control gene be explained?

3) Also, at line 164, the authors comment that NRF2 is suppressed in Covid-19 patients, which contradicts the Figure 4 data that shows NRF2 expression seems to be higher mainly in Moderate COVID-19 and pre-CAM disease comparing to health subjects even using the CypA as the internal control. Thus, the authors should specify the difference in the expression profile of NRF2 inter-group based on the usage of CypA.

Minor comments:

- 1) The official name of SARS-CoV-2 disease is COVID-19.
- 2) Figure 1 and Figure 2 are distorted.
- 3) Line 223-224: "significant up-regulation of NRF2 was observed in mild (>250 fold), asymptomatic (>15 fold), moderate (>12 fold)". The authors should specify which disease and the figure number related to these data.
- 4) Line 110: change SARS CoV-2 by SARS-CoV-2
- 5) Line 152-155: The authors should specify the figure number related to the data (Figure 3e).
- 6) Line 214-215: The authors should specify the table number related to the data (Table 2).

Staff Comments:

Preparing Revision Guidelines

Please return the manuscript within 60 days; if you cannot complete the modification within this time period, please contact me. If you do not wish to modify the manuscript and prefer to submit it to another journal, please notify me of your decision immediately so that the manuscript may be formally withdrawn from consideration by Microbiology Spectrum.

Comments

The manuscript by Sunil et al., has carried out very important study during this ongoing COVID-19 pandemic. Although, this study provides strong rationale for identification of appropriate reference genes in the viral infections but has some serious problems in the experiments and result interpretations. Overall, the results are not sufficient and convincing to their claims. Here are my comments for the authors.

1. The authors should show the viral genome content in the subjects analyzed.
2. I think it is difficult to interpretate the results in this kind of experimental settings since the reference gene expression levels can be affected by several factors not only due to SARS-CoV-2 infections. For example, the age and health conditions of the subjects in the groups can be different.
3. Authors did not mention the quantity of cDNA was used in the RT-PCR reactions.
4. It was mentioned that the groups were categorized by institutional definitions, but the categorization of the groups must be approved by certain international bodies otherwise, it is hard to apply the findings of this study in general.
5. In Fig.2, some of the ct values for the GUSB or PGC-1a or GAPDH in pre-CAM or post-CAM are above ct 30.0, in some cases even above ct 34.0. I think the ct values above 30 is not reliable for calculations. Moreover, the PCR was performed at annealing temperature of 60 degrees for all the reference genes. However, the primers could have better amplification efficiency at other temperatures as well. Not sure, if authors have performed gradient PCR and selected the annealing temperature for the primers.
6. A 260/280 ratio of ~2.0 is generally accepted as “pure” for RNA. In Table 1, the RNA quality for the groups of Severe, pre-CAM and post-CAM is below 2.0 and varies a lot compared to other groups. This is very important factor in the whole experiment as the downstream steps depends on the purity of RNA used for cDNA preparation.
7. To support the RT-qPCR data, authors should have analyzed the expression of these reference genes by Western blotting from the PBMC's collected.
8. Based on normalization with CypA, NRF2 expression level did not correlate with the disease severity in the groups. Therefore, authors should have analyzed for at least 4-5 genes that may have differential expression in SARS-CoV-2 infections to prove that CypA is the suitable reference gene compared to GAPDH.

Dear Dr. Toth,

At the outset, we would like to thank the reviewers for taking out time and providing valuable comments that has allowed us to further strengthen the manuscript. In the light of these comments, we have performed new experiments including gene expression of two additional target genes (*IL-6* and *IL-15*), supporting cytokine (IL-6) and protein quantification data (CRP), and rephrased interpretations in the light of new data that has further strengthened our conclusions. Therefore, we believe that in the revised manuscript, we have adequately addressed the concerns raised by the reviewers. A point-by-point response is attached below:

Reviewer #1

Comment 1: The authors should show the viral genome content in the subjects analysed.

Response: We thank the reviewer for this comment. Actually, all the COVID-19 patients in this study were confirmed for the presence of viral genome; and not by antigen or any other test. The revised manuscript now has the following statement in the material and methods (lines **351-55**, revised manuscript):

“The COVID-19 status of the subjects was confirmed by the presence of viral content using Xpert Xpress SARS-CoV-2 (Cepheid, USA) or TrueNat system (Molbio Diagnostics, India) approved by the Indian Council of Medical research for COVID-19 diagnostics. Briefly, the patients were evaluated for the presence of *N2* and *E* gene of SARS-CoV-2 (SARS-CoV-2 Xpert). In case of TrueNat, initial screening was based on the presence of *E* gene; followed by *RdRp* gene in case results of screening are positive.”

Furthermore, the sub-categorization of the patients into various categories (mild- severe) in our study was as per the guidelines of AIIMS/Indian Council of Medical Research, Ministry of Health, Govt. of India. These classifications and guidelines are universal in nature and are consistent with the guidelines of other global bodies such as NIH, USA and WHO. Broadly, these guidelines are based on the well-established premise that clinical severity is primarily manifested based on host response to infection independent of viral load.

1. To, K. K., Tsang, *et al* (2020). Temporal profiles of viral load in posterior oropharyngeal saliva samples and serum antibody responses during infection by SARS-CoV-2: an observational cohort study. *The Lancet. Infectious diseases*, 20(5), 565–574.
2. Zou L *et al.*, (2020) Viral Load in Upper Respiratory Specimens of Infected Patients. *N Engl J Med*. 19; 382 (12):1177-1179.
3. Arons, M. M., *et al* and CDC COVID-19 Investigation Team (2020). Pre-symptomatic SARS-CoV-2 Infections and Transmission in a Skilled Nursing Facility. *The New England journal of medicine*, 382(22), 2081–2090.

4. Singanayagam, A et al, (2020). Duration of infectiousness and correlation with RT-PCR cycle threshold values in cases of COVID-19, England, and January to May 2020. *Euro surveillance: European communicable disease bulletin*, 25(32), 2001483.
5. Jacot, D., Greub, G., Jaton, K., & Opota, O. (2020). Viral load of SARS-CoV-2 across patients and compared to other respiratory viruses. *Microbes and infection*, 22(10), 617–621.

We have now added statements in the revised manuscript alluding to these facts (Lines **251-53**; and Lines **351-55**).

Comment 2: I think it is difficult to interpretate the results in this kind of experimental settings since the reference gene expression levels can be affected by several factors not only due to SARS-CoV-2 infections. For example, the age and health conditions of the subjects in the groups can be different.

Response: We don't understand "this kind of experimental" settings as this work involves patients' samples, and the settings are as good as for any clinical study. The subjects in our data set represent **the real-world patient heterogeneity** which is not adequately captured by the standard laboratory experimental set-up including with the animal studies. The whole premise of the study was to select an appropriate reference gene that could capture precisely these kind of broad-spectrum manifestations in the patients having COVID-19 infection (asymptomatic, symptomatic, mild, moderate and severe) including co-infection of mucormycosis. The very definition of the ideal reference gene warrants that the gene expression should be minimally regulated across the disease spectrum so as to have minimal influence of the patient heterogeneity in terms of age/sex or severity, treatment etc. It is one of the primary rationales to experimentally determine an appropriate reference gene.

In this context, we screened nine popularly used reference genes as candidates and determined how their expression vary across patient groups. These patient groups were manifesting different clinical symptoms and had a broader age profile (Table 3 original manuscript) even though the **mean age** of the patients was similar across the groups (Figure 1, rebuttal).

Figure 1: Mean age of patients enrolled in particular group for this study. Graph represent Mean±SEM. One-way ANOVA was used to ascertain significance and no significant difference exist among various groups.

These patients received different treatments as per the clinical requirement ascertained by attending physician based on risk and severity (asymptomatic, symptomatic, mild, moderate and severe), and co-infection of mucormycosis. As per the human ethical approval guidelines, we have clearly stated in **line no 383** that “The design of study did not influence the clinical care or treatment for any of the patient.”

Indeed, a large number of the nine candidate reference genes displayed variations in gene expression based on patient heterogeneity as was expected. That is why those genes did not satisfy the prerequisites of a suitable reference gene. Overall, only *CypA* demonstrated consistent expression across all the patient groups irrespective of any possible clinical confounder and accordingly emerged as the most suitable reference gene in this study. Our work is therefore consistent with the guidelines to reference gene selection for quantitative real-time PCR (Radonić *et al*, 2004) as the reference genes identified in our study have expression prevalence across the PBMCs of various categories of patients including that of co-infection, and are minimally regulated allowing the accuracy of RNA transcription analysis. The study design also adheres to the established standards of clinical research including the one used to determine reference genes in both humans and animals (Dheda *et al*, 2004; Montero-Melendez & Perretti, 2014; Pilbrow *et al*, 2008; Silver *et al*, 2006).

Based on the scientific arguments presented above, we believe that our study truly captures the real clinical scenario and highlights the heterogeneity of expression in popular/commonly used reference genes; and finally identifies *CypA* as an ideal reference gene whose expression remains consistent across the clinical spectrum. Wherever necessary we have now expanded on our discussion emphasizing on these facts. We thank the reviewer for the comment as it allowed us to emphasize more clearly on the limitations of conventional reference genes and suitability of *CypA* as reference gene.

Comment 3: Authors did not mention the quantity of cDNA was used in the RT-PCR reactions.

Response: We apologise for this oversight. The final concentration of cDNA was 10 ng/reaction in qRT-PCR and the same is now included in the revised manuscript (Line no 416).

Comment 4: It was mentioned that the groups were categorized by institutional definitions, but the categorization of the groups must be approved by certain international bodies otherwise, it is hard to apply the findings of this study in general.

Response: Thank you for highlighting this important point. AIIMS, New Delhi (Our institute) was the National Nodal Centre for COVID-19 and follows the same guidelines as recommended by Indian Council of Medical Research, ICMR, and Govt. of India.

These guidelines are also similar to guidelines from other international organisations such as WHO, and NIH, USA. In the revised work, we have appropriately rephrased and expanded on these guidelines (lines no **363-83**).

Comment 5: In Fig.2, some of the ct values for the GUSB or PGC-1a or GAPDH in pre-CAM or post-CAM are above ct 30.0, in some cases even above ct 34.0. I think the ct values above 30 is not reliable for calculations. Moreover, the PCR was performed at annealing temperature of 60 degrees for all the reference genes. However, the primers could have better amplification efficiency at other temperatures as well. Not sure, if authors have performed gradient PCR and selected the annealing temperature for the primers.

Response: Thank you. We respond to this comment in two parts as indicated below:

- a) **Some of the ct values for the GUSB or PGC-1a or GAPDH in pre-CAM or post-CAM are above ct 30.0, in some cases even above ct 34.0. I think the ct values above 30 is not reliable for calculations.**

We appreciate the reviewer's comment. First, we respectfully disagree with the subjective opinion that values above Ct 30 are not reliable. This assertion belies scientific evidence, and contradicts the well-established and universally accepted scientific premises/principals. Essentially, Ct values represent transcriptional level of a gene in that particular cell/condition; if any gene is less represented/less abundant, that will be reflected in the Ct values. Based on the natural expression, the level may be less or high which will be reflected in the Ct value. Ct values actually are considered in acceptable range until it does not cross the limit of detection or negative control/NTC (Non-template control) and rather than arbitrary cut-off values as indicated in the comment.

Guidelines to reference gene selection for quantitative real-time PCR recommends a Ct ≤ 40 for determination of qRT-PCR efficiency and intra- and inter-assay variability for selection of appropriate gene (Radonić *et al.*, 2004). Consequently, literature is replete with the studies supporting the Ct values above 30 for scientific interpretations and experimentations including confirming of clinical status and follow-up care (Dhedra *et al.*, 2004; Radonić *et al.*, 2004; Silver *et al.*, 2006). Heterogeneity in gene expression among patients is expected, and commonly present, and same is aptly reflected in the Ct values. It is precisely this heterogeneity that makes it essential to determine the most suitable reference gene, especially when working with patients. We have adequately highlighted these facts by providing ranges of Ct values observed in our study (Lines 108, lines 119-124, original manuscript).

In our manuscript, the differences in Ct values of a specific reference gene from a particular group that the reviewer is alluding to are in fact natural variations in expression observed among patients (Brym *et al.*, 2013; Gao *et al.*, 2008; Silver *et al.*, 2006; Yu *et al.*, 2008). It is well reported that these housekeeping genes are involved in plethora of cellular pathways, which may get dysregulated in patients based on variety of factors including severity (Dhedra *et al.*, 2004; Montero-Melendez & Perretti, 2014; Shivashankar *et al.*, 2021). Further, most qRT-PCR based COVID-19 diagnostics kits approved by WHO, CDC and other National and international agencies consider acceptable range is \sim Ct ≤ 40 (Table 1, rebuttal). In our study, irrespective of variations in severity of infection, the overall Ct values remain within the

range of each group (dispersal range 6.43- 39.69; lines 108 original manuscript) and we could successfully identify the most stable reference gene to be used during COVID-19.

Table 1: National and International approved kits for COVID-19 diagnostics with the acceptable range $C_t \geq 35$

S.No.	Name of Kit	Manufacturer	C_t cut off	Publihsed studies using these diagnostics kits
1	MOLgen SARS-CoV-2 Real Time RT-PCR Kit	Adaltis	$C_t \leq 40$	1. Farhana, N., Choudhury, R., & Khanam, F. (2022). Detection of SARS-CoV2 by a Commercial RNA Detection Kit: a Public Health Laboratory Experience. International Journal of Infectious Diseases, 116, S42. 2. Favaro, M., Mattina, W., Pistoia, E. S., Gaziano, R., Di Francesco, P., Middleton, S., ... & Fontana, C. (2021). A new qualitative RT-PCR assay detecting SARS-CoV-2. Scientific reports, 11(1), 1-9.
2	Allplex 2019-nCoV assay	Seegene	$C_t < 40$	1. Cotugno, N., Ruggiero, A., Bonfante, F., Petrara, M. R., Zicari, S., Pascucci, G. R., Zangari, P., De Ioris, M. A., Santilli, V., Manno, E. C., Amodio, D., Bortolami, A., Pagliari, M., Concato, C., Linardos, G., Campana, A., Donà, D., Giaquinto, C., CACTUS Study Team, Brodin, P., ... Palma, P. (2021). Virological and immunological features of SARS-CoV-2-infected children who develop neutralizing antibodies. Cell reports, 34(11), 108852. 2. Farfour, E., Lesprit, P., Visseaux, B., Pascreau, T., Jolly, E., Houhou, N., ... & Vasse, M. (2020). The Allplex 2019-nCoV (Seegene) assay: which performances are for SARS-CoV-2 infection diagnosis?. European Journal of Clinical Microbiology & Infectious Diseases, 39(10), 1997-2000. 3. Freppel, W., Merindol, N., Rallu, F., & Bergevin, M. (2020). Efficient SARS-CoV-2

				detection in unextracted oro-nasopharyngeal specimens by rRT-PCR with the Seegene Allplex™ 2019-nCoV assay. Virology Journal , 17(1), 1-10.
3	TaqPath COVID-19 Combo Kit	Thermo Fisher, USA	Ct < 40	1. Welch, Nicole L., et al. "Multiplexed CRISPR-based microfluidic platform for clinical testing of respiratory viruses and identification of SARS-CoV-2 variants." Nature medicine 28.5 (2022): 1083-1094. 2. Uribe-Alvarez, Cristina, et al. "Low saliva pH can yield false positives results in simple RT-LAMP-based SARS-CoV-2 diagnostic tests." PloS One 16.5 (2021): e0250202.
4	Lab Gun Real-Time PCR Kit	Lab Genomics	Ct < 40	1. Garg, A., Ghoshal, U., Patel, S. S., Singh, D. V., Arya, A. K., Vasanth, S., ... & Srivastava, N. (2021). Evaluation of seven commercial RT-PCR kits for COVID-19 testing in pooled clinical specimens. Journal of medical virology , 93(4), 2281-2286.
5	Real-Time Fluorescent RTPCR Kit for 2019-nCoV	BGI Genomics	Ct < 37	1. Khan, R. S., & Rehman, I. U. (2020). Spectroscopy as a tool for detection and monitoring of Coronavirus (COVID-19). Expert review of molecular diagnostics, 20(7), 647-649. 2. Okwurawe, A. P., Onwuamah, C. K., Shaibu, J. O., Amoo, S. O., Ige, F. A., James, A. B., ... & Audu, R. A. (2021). Low level SARS-CoV-2 RNA detected in plasma samples from a cohort of Nigerians: Implications for blood transfusion. PloS One, 16(6), e0252611.

b) The PCR was performed at annealing temperature of 60 degrees for all the reference genes. However, the primers could have better amplification efficiency at other temperatures as well. Not sure, if authors have performed gradient PCR and selected the annealing temperature for the primers.

Thank you. Firstly, Figure 1 of our original manuscript clearly show a single melt curve and melt peak for each individual gene. This is indicative of a single amplicon denoting specificity of the primers at the conditions used. Further, primer designing was performed in a manner to have their expected best amplification efficiency at 60°C; and it was validated using healthy subjects. Nonetheless, to address reviewer's concern, we again checked

amplification efficiency of primers of individual genes through gradient PCR by selecting a temperature range of 58°C-61°C and with three different subjects (Figure 2, rebuttal).

As shown below, we observed the best amplification efficiency for primers for each candidate reference gene at 60°C only.

Figure 2: Gradient PCR for the determination of optimum annealing temperature to obtain the maximum amplification efficiency of primers for candidate reference genes. Data show the gene amplicons obtained with their respective primers at different annealing temperature (58°C, 59°C, 60°C, and 61°C). As evident from the data, annealing temperature of 60°C yielded the maximum amplification as highlighted by the band intensity. Sample 1-3 represent three different healthy subjects; X-axis denotes temperature range used in the gradient-PCR. Left panel (Y-axis) lists the genes used in this study. This figure is now added as supplemental Fig. S1 in revised manuscript.

Comment 6: A 260/280 ratio of ~2.0 is generally accepted as "pure" for RNA. In Table 1, the RNA quality for the groups of Severe, pre-CAM and post-CAM is below 2.0 and varies a lot compared to other groups. This is very important factor in the whole experiment as the downstream steps depends on the purity of RNA used for cDNA preparation.

Response: We understand the reviewer’s concern. 260/280 ratios actually can be impacted by several factors including the complexity of clinical samples, storage, template composition, and even solution used to suspend RNA (Okamoto & Okabe, 2000; Willinger *et al*, 2017; Yu *et al.*, 2008). While 260/280 ratio of ~2.0 is desirable, the 260/280 values ~1.7 are acceptable and frequently used (Ghawana *et al*, 2011; Willinger *et al.*, 2017; Zhang & Yang, 2022). In fact, depending on the complexity of samples including clinical samples, DNA/RNA 260/280 ratios <1.6 are considered good for any downstream applications including the development of biomarkers (Banki *et al*, 2020; Liu *et al*, 2021; Seres-Bokor *et al*, 2021; Tantoh *et al*, 2019; Willinger *et al.*, 2017; Yu *et al.*, 2008).

In our study, comparison of 260/280 ratio across all groups revealed that there were no significant differences in the mean 260/280 ratio among various groups including the severe COVID-19 and CAM groups (Figure 3, rebuttal).

Figure 3: Mean 260/280 ratio of RNA isolated from various subjects used in the study.

There was no significant difference in the 260/280 ratio (Y-axis) across the groups (X-axis). Data represent mean±SD; One-Way ANOVA with Kruskal-Wallis test with post-hoc corrections. “ns” represents non-significant differences (*, *P*-value ≤0.05). This figure is now added as supplemental Fig. S2 in revised manuscript.

The mean 260/280 ratio was in the range of 1.94 to 1.98) for healthy, asymptomatic and mild infected; and (1.73 to 1.79) in case of CAM and severe COVID-19 patients. Incidentally, the mortality rates among these cohorts of severe COVID-19 and CAM were over 30%; and ~50%, respectively. Nonetheless, to rule out the effect of these- minor and statistically non-significant variations in 260/280 ratio among various groups, we re-analyzed our data by excluding patient data with 260/280 ratio <1.7, we observe that this did not alter the conclusions of our analysis and *CypA* remained the most suitable reference gene.

In the revised manuscript, we have added points pertaining to this discussion (Lines no 292-99)

Comment 7: To support the RT-qPCR data, authors should have analysed the expression of these reference genes by Western blotting from the PBMC's collected.

Response: We respectfully disagree. The aim of our study was to find the most suitable reference gene for qRT-PCR data analysis in COVID-19 and CAM patients across the disease spectrum. It is well known that qRT-PCR data measures expression of gene (transcription), while the WB analysis incorporates additional elements including mRNA half-life, translational efficiency, trafficking, stability and post-translational modifications.

Furthermore, SARS-CoV-2 is known to affect splicing, translation and trafficking of host proteins to suppress host response (Banerjee *et al*, 2020; Finkel *et al*, 2021). During infection, SARS-CoV-2 develops multiple strategies like degradation of host mRNA pool to hijack host translation machinery to synthesize its own protein (Finkel *et al.*, 2021; Fisher *et al*, 2022). Considering these well-established premises, it is not clear to us how WB analysis would have supported meaningful interpretation of qRT-PCR data. This fact is also clearly reflected in the similar studies pertaining to identification of suitable reference gene for qRT-PCR data analysis in other organisms/conditions (Table 2, rebuttal), none of which have performed WB analysis.

Table 2: Studies pertaining to identification of suitable reference gene for qRT-PCR data analysis in other organisms/conditions. None of these studies have employed WB data to support conclusions of qRT-PCR data.

Sr. No	Studies involving selection of reference genes across various organisms including humans
1	Silver, Nicholas, et al. (2006) "Selection of housekeeping genes for gene expression studies in human reticulocytes using real-time PCR." BMC molecular biology 7.1: 1-9.
2	Dheda, Keertan, et al. (2004) "Validation of housekeeping genes for normalizing RNA expression in real-time PCR." Biotechniques 37.1: 112-119.
3	Kidd, Mark, et al. (2007) "GeneChip, geNorm, and gastrointestinal tumors: novel reference genes for real-time PCR." Physiological genomics 30.3: 363-370.
4	Kuang, Jujiao, et al. (2018) "An overview of technical considerations when using quantitative real-time PCR analysis of gene expression in human exercise research." PloS One 13.5: e0196438.
5	Mehta, Rohini, et al. (2010) "Validation of endogenous reference genes for qRT-PCR analysis of human visceral adipose samples." BMC Molecular Biology 11.1: 1-10.
6	Aggarwal, Anu, et al. (2018) "Optimal reference gene selection for expression studies in human reticulocytes." The Journal of Molecular Diagnostics 20.3: 326-333.
7	Augustyniak, Justyna, et al. (2019) "Reference gene validation via RT-qPCR for human iPSC-derived neural stem cells and neural progenitors." Molecular Neurobiology 56.10: 6820-6832.

Nonetheless, in the revised manuscript, we have now determined the *IL-6* gene expression using normalization with *CypA*, and *GAPDH*. We correlated gene expression profile with the levels of cytokine IL-6 in these patients. We demonstrate that gene expression values for *IL-6* obtained by normalization with *CypA*, and not with *GAPDH*, are consistent with the IL-6 levels in plasma. These new data and discussion are now added in the revised manuscript (Lines no **203-216; and 269-71**).

Comment 8: Based on normalization with CypA, NRF2 expression level did not correlate with the disease severity in the groups. Therefore, authors should have analysed for at least 4-5 genes that may have differential expression in SARS-CoV-2 infections to prove that CypA is the suitable reference gene compared to GAPDH.

Response: We thank the reviewer for this insightful comment. To confirm the findings of *CypA* based normalization *vis a vis* *GAPDH* based normalization, we wanted to use the genes whose expression is previously confirmed in COVID-19 patients using RNA sequencing. This would allow us to evaluate how closely qRT-PCR data normalized using *CypA* or *GAPDH* compares with the RNA sequencing-based studies. It is in this context that we used major redox regulator *NRF2* whose expression is dysregulated during COVID-19, and is not sufficiently up-regulated, and remained relatively suppressed/minimally induced to mount an effective anti-oxidant response (Zhang *et al*, 2022). During oxidative stress conditions such

as inflammation, *NRF2* is usually significantly induced to maintain redox homeostasis. The relative suppression of *NRF2* expression by SRAS-CoV-2 is a viral strategy to hijack host protective responses (Olagnier *et al.*, 2020; Zhang *et al.*, 2022) and induction of *NRF2* pharmacologically is proposed to be an effective strategy against COVID-19 (McCord *et al.*, 2021).

We first show that our data on *NRF2* expression based on *CypA*-normalization is actually consistent with the previous reports on *NRF2* expression showing that SARS-CoV-2 infection dysregulates *NRF2* to impair its proper induction leading to an increased oxidative stress, a hallmark feature of COVID-19 (Olagnier *et al.*, 2020; Zhang *et al.*, 2022). Similar to RNA sequencing-based study, normalization with *CypA* revealed a relatively low *NRF2* expression even in the oxidative stress environment of COVID-19-associated inflammation. Moreover, we did not observe significant difference in the expression of *NRF2* across the spectrum of COVID-19 infection (Figure 4, rebuttal and Fig. 4 in revised manuscript), as is also indicated from RNA Sequencing analysis (Jain *et al.*, 2021). On the other hand, normalization with *GAPDH* in our study rather indicated massive up-regulation of *NRF2* expression except in case of severe COVID-19 patients.

Figure 4: Gene expression profile of *NRF2* based on use of *CypA* or *GAPDH* as internal control.

NRF2 expression based on *CypA*-normalization reveals no significant difference exists in its expression across various patients' groups. This is consistent with the RNA sequencing-based expression wherein *NRF2* expression is relatively low even in the high oxidative stress environment reported for COVID-19 and CAM infections (Jain *et al.*, 2021; Olagnier *et al.*, 2020). On the other hand, normalization with *GAPDH* reveals significant expression heterogeneity that varies from 2-fold to over 600-fold across spectrum of COVID-19 and CAM infections. This does not match with RNA sequencing-based data from published works and erroneously creates the impression of *NRF2* over-expression. Healthy subjects, represents the baseline *NRF2* expression. Data represents mean \pm standard error of the mean (SEM) performed with at least eight subjects in duplicates. Significance is ascertained by using a One-way ANOVA with Kruskal-Wallis test with Dunn's post-hoc corrections, N=8/group (*, *P*-value < 0.05; **, *P*-value < 0.01; ***, *P*-value < 0.001, and ns=non-significant).

For further strengthening of our conclusions about utility of *CypA*, in the revised work, we have additionally determined the expression of *IL-6* and *IL-15* genes (Figure 5 and 6 of rebuttal, and Fig. 5-7 in revised manuscript). Furthermore, the data on *IL-6* gene expression,

a marker for inflammation is also correlated with cytokine IL-6 levels. The inflammation status as suggested by IL-6 is further corroborated by determining CRP levels.

IL-6 gene expression: *IL-6* expression is regulated by *NRF2* as *NRF2* blocks the pro-inflammatory responses by inhibiting the expression of *IL-6* at the transcription level (Kobayashi *et al*, 2016; Matsuoka *et al*, 2016). A negative relationship therefore exists between expression of *NRF2* and *IL-6*, and a high *NRF2* expression should lead to suppression of *IL-6* expression. Surprisingly, we observe that the normalization with *GAPDH* as reference gene produces a high *NRF2*–high *IL-6* expression phenotype as both *IL-6* and *NRF2* showed massive induction (>400 fold) in case of patients with mild, and CAM (pre and post) infections. This shows that *GAPDH* is not an ideal reference gene for qRT-PCR-based normalization during COVID-19. Moreover, we also did not observe any significant differences in the expression of *IL-6* between asymptomatic, and severely infected COVID-19 patients when data is normalized with *GAPDH*. This is again in contrast to the established fact that inflammation is minimal in asymptomatic, and very high in patients having severe COVID-19.

Figure 5: Gene expression profile of *IL-6* based on normalization with *CypA* or *GAPDH* as internal control gene. With *GAPDH*, we observed similar expression among asymptomatic, mild, moderate and severe infections. On the other hand, RNA sequencing data indicates a severity dependent increase in *IL-6* expression going from mild to severe infections (Mukhopadhyay *et al.*, 2021; Islam *et al.*, 2021; Delgado-Roche *et al.*, 2020). Only *IL-6* expression derived from normalization with *CypA*, therefore, is in agreement with previously published studies. Y-axis represents differences in the fold-change based on data normalization *vis à vis* *CypA* or *GAPDH* in various groups (X-axis) of COVID-19 and CAM patients. Significance is ascertained with respect to healthy subjects by using a One-way ANOVA with Kruskal-Wallis test with Dunn’s post-hoc corrections (*, *P*-value <0.05; **, *P*-value < 0.01; ***, *P*-value < 0.001, and ns=non-significant).

On the other hand, normalization of data with *CypA* showed an overall higher *IL-6* and low *NRF2* expression in all groups consistent with the established relationship of *IL-6* and *NRF2* expression. The highest *IL-6* expression was observed in severe COVID-19 patients that had the lowest *NRF2* expression. Furthermore, expression of *IL-6* in asymptomatic, mild and severe cases differed significantly, as it should be as inflammation levels vary significantly among these groups with asymptomatic having the lowest and severe having the highest inflammation.

To further support our observations, we next determined **IL-6** levels in plasma (Figure 6, rebuttal, and Fig. 6a in revised manuscript) and noticed that compare to *GAPDH*, the *CypA* normalized *IL-6* expression concurs well with the IL-6 levels further validating *CypA* as a reference gene. Asymptomatic patients have IL-6 levels comparable to healthy individuals. However, *GAPDH* normalized data showed non-significant differences in *IL-6* expression between severe and asymptomatic.

Figure 6: IL-6 level in plasma across the spectrum of COVID-19 severity and CAM. IL-6 level (Y-axis) reveals significant differences across the patients in different groups (X- axis) of COVID-19 spectrum and CAM. Asymptomatic patients have the lowest and patients having severe COVID-19 have the highest IL-6 level. These data support the gene expression profile derived from normalization with *CypA*. Significance is ascertained by using a One-way ANOVA with Kruskal-Wallis test with post-hoc corrections. N=8-12/group (*, P -value <0.05; **, P -value <0.01; ***, and P -value <0.001).

Estimation of CRP level: To additionally strengthen our conclusions, we determined plasma CRP (C-reactive protein) levels to reflect on the status of systemic inflammation among these patients. We observed a direct correlation between CRP and disease severity, with asymptomatic subjects having the lowest, and severe COVID-19 patients having the highest CRP levels (Figure 7 in rebuttal, and Fig. 6b in revised manuscript). We observed that this data concurs with the plasma IL-6 levels, and *IL-6* gene expression when *CypA* is used for normalization.

Figure 7: C-reactive protein (CRP) level in plasma across the spectrum of COVID-19 severity and CAM. CRP level (Y-axis) reveals significant differences across the patients in different groups (X- axis) of COVID-19 spectrum and CAM. Asymptomatic patients have the lowest and patients having severe COVID-19 have the highest CRP levels. These data matches with the gene expression profile derived from normalization with *CypA*.

Significance is ascertained by using a One-Way ANOVA with Kruskal-Wallis test with post-hoc corrections. N=8-12/group (*, P -value < 0.05; **, P -value < 0.01; ***, and P -value < 0.001).

***IL-15* gene expression:** In addition, we also scored for the gene expression of *IL-15* and observe that the expression profile obtained following normalization with *CypA* corresponds better with the RNA sequencing based published work.

Figure 8: Gene expression profile of *IL-15* gene based on the usage of *GAPDH* or *CypA* as the internal control gene. Normalization with *GAPDH* does not concur with RNA-sequencing-based data which shows that *IL-15* expression increases significantly from mild to moderate, and then returns to baseline expression in severely sick (Jain et al., 2021). In case of normalization with *CypA*, expression pattern concurs with RNA sequencing data by demonstrating an increase from mild to moderate, and a reduction in severely sick. Y-axis represents differences in the fold-change based on data normalization in various groups (X-axis) of COVID-19 and CAM. Significance is ascertained by using a One-way ANOVA with Kruskal-Wallis test with Dunn's post-hoc corrections (*, P -value < 0.05; **, P -value < 0.01; ***, and P -value < 0.001).

Taken together, our additional data involving expression profiling of genes namely *IL-6*, and *IL-15* along with determination of plasma levels of IL-6 and CRP across the spectrum of COVID-19 and mucormycosis infections firmly establish the utility of *CypA* as the reference gene.

The statements alluding to the above facts and additional data have now been added to the revised manuscript (lines no 190-221 and 302-336).

Reviewer #2 (Comments for the Author)

The manuscript by Kumar et al., entitled "Selection of ideal reference genes for gene expression analysis in COVID-19, and Mucormycosis infection" identified among 9 different housekeeping genes, the *CypA* gene as the most suitable reference gene, which accurately captured the heterogeneity of infection and yielded a stable expression across different conditions of COVID-19 severity and Mucormycosis infection (Covid-associated mucormycosis - CAM). Briefly, the authors tested the expression stability of nine different housekeeping genes (including TBP, *CypA*, B2M, 18S, PGC-1 α , GUSB, HPRT-1, β -ACTIN and *GAPDH*) in PBMC isolated from patients with different COVID-19 severity and CAM. The analysis of housekeeping genes during SARS-CoV-2 spectrum disease and CAM infection showed significant expression variations in most of candidate reference genes intra and inter-group. Only TBP, *CypA*, B2M and HPRT-1

had a comparable mean Ct value across SARS-CoV-2 spectrum and CAM infection. Then, the authors employed statistical algorithms to determine the stability indexes of each reference gene. GHPDH and PGC-1 α had more deviations. Nonetheless, CypA is identified as the most suitable reference gene in Covid-19, and CAM infections.

Strength

Studying the co-infection of SARS-CoV-2 and mucormycosis is relevant based on existing public health issue and the lack of studies published in the field. In addition, the selection of accurate reference genes in different disease outcomes could advance the field.

Weaknesses

The observations postulated in the paper are novel and interesting. However, the discussion is not robust enough to make convincing conclusion. The authors should show more data from published studies related to their results.

Response:

We thank the reviewer for the encouraging words and highlighting strength and weaknesses. In the revised manuscript, we have now significantly strengthened our conclusions by providing additional data, including more candidate genes for validation of *CypA* as reference gene and supplementing this data by quantifying levels of CRP and IL-6 in the plasma of the patients. In the light of additional data, and a significantly improved discussion, we believe that we have now adequately addressed the comments in the revised manuscript. We thank the reviewers for their time and efforts to improve our work.

Major comments

Comment 1: Figure 1- Figure 3: Figure 2 shows difference between healthy and Covid19 subjects with moderate or severe or CAM infections. GAPDH showed variations inter-groups, in which expression was comparable in healthy, mild and asymptomatic patients, while moderate and asymptomatic as well as Pre/Pos CAM had other expression patterns. How could you explain this pattern of high GAPDH variations inter-group (Figure 2) whether in Figure 3b the data shows GAPDH as a reference gene more stable than CypA?

Response: We appreciate the reviewer for this comment and apologise for this confusion, which has arisen primarily due to Figure 3b. Following the reviewer's comment, we revisited this figure and realised an inadvertent error in the processing of Bestkeeper data that factored in only coefficient of correlation (r^2) without incorporation of the stability values. Now, we have fixed this oversight and reprocessed this data (**Fig. 3b in revised manuscript, and Figure 9 of rebuttal**), where *CypA* showed highest stability value whereas *GAPDH* showed least stability value. Therefore, the stability value of *CypA* is now in accordance to Figure 2 of original manuscript. We thank the reviewer for bringing this to our notice that has allowed us to fix it.

Figure 9: Comparison of candidate reference genes based on cumulative ranking score using different statistical algorithms.

(a) Intra-group average standard deviations for individual patient groups were obtained using delta Ct method, and are added to obtain a cumulative average standard deviation (Y-axis). The highest cumulative average standard deviation (i.e., *GAPDH*) indicates the least gene stability (X-axis). (b) The graph represents a cumulative stability value (Y-axis) obtained by adding intra-group stability value based on coefficient of correlation (r) and standard deviation using Bestkeeper algorithm. The candidate gene yielding the lowest cumulative value (i.e., *PGC-1 α*) indicates the highest stability. (c) In Normfinder algorithm, intra-group stability value (Y-axis) based on coefficient of variations are added to obtain a cumulative stability value. The candidate gene yielding lowest cumulative stability value (i.e., *CypA*) indicates the most stable reference gene. (d) In GeNorm algorithm, intra-group expression stability (M value) based on the average pairwise variation of an individual gene with all other genes used as control to obtain a cumulative expression stability (Y-axis). The gene yielding the highest M value (i.e., *GAPDH*) indicates the lowest stability. (e) In RefFinder algorithm, intra-group geometric mean (Y-axis) based on geometric mean of ranking values calculated by all above algorithms are added to obtain a cumulative geometric mean of ranking values. *CypA* yielded the lowest cumulative geometric mean indicating the highest stability and therefore emerged as the most suitable reference gene.

Furthermore, *GAPDH* is a multi-faceted gene, which in addition to glycolysis, is also involved in diverse cellular functions including regulation of inflammation (Montero-Melendez & Perretti, 2014). Inter-group variability in *GAPDH* expression can also be due to the differential inflammation status observed in COVID-19 patients as is also highlighted by IL-6 and CRP data presented in the revised manuscript (Millet *et al*, 2016; Shivashankar *et al.*, 2021). We have now expanded these statements in the discussion section of the revised manuscript (Lines no- 268-271).

Comment 2: Figure 4: As the health subjects represent the baseline NRF2 expression, is there significant difference in NRF2 expression among samples from moderate, asymptomatic and/or pre/post-CAM diseases when *CypA* was used as internal control gene? If so, how can the NRF2 expression variability among them using *CypA* as internal control gene be explained?

Response: We appreciate the reviewer's comment. In the revised manuscript, we have performed the statistical analysis with the data of *NRF2* gene expression from healthy subjects as baseline. Even though there are differences in expression of *NRF2* across various categories of patients, these differences are not statistically significant when data is normalized with *CypA* as internal control. This is consistent with the earlier reports in COVID-19 where in *NRF2* expression is relatively suppressed/minimally induced even though the inflammation levels and consequent oxidative stress levels are very high (Olagnier *et al.*, 2020; Zhang *et al.*, 2022).

NRF2 is a major regulator of host anti-oxidant response, which also regulates the transcription of cytoprotective genes to mitigate the oxidative stress mediated tissue damage, and inflammatory response induced by viral infection. Nonetheless, during COVID-19, *NRF2* expression is dysregulated and not sufficiently induced to mount an effective anti-oxidant response (Zhang *et al.*, 2022). The relative suppression of *NRF2* expression by SRAS-CoV-2 is a viral strategy to hijack host protective responses (Olagnier *et al.*, 2020; Zhang *et al.*, 2022) and induction of *NRF2* expression is proposed to be an effective strategy against COVID-19 (McCord *et al.*, 2021). The slightly higher *NRF2* levels observed in CAM group can be a response to mucormycosis infection, and *NRF2* expression returns closer to base-line following reduction in fungal burden post-surgery (post-CAM).

We have alluded to these facts in the revised manuscript. **Also, please refer to our response to comment 8, reviewer 1.**

Comment 3: Also, at line 164, the authors comment that NRF2 is suppressed in Covid-19 patients, which contradicts the Figure 4 data that shows NRF2 expression seems to be higher mainly in Moderate COVID-19 and pre-CAM disease comparing to health subjects even using the CypA as the internal control. Thus, the authors should specify the difference in the expression profile of NRF2 inter-group based on the usage of CypA.

Response: As explained above, we have used *NRF2* only as a candidate gene whose expression was previously reported in COVID-19 patients using RNA sequencing. We now provide data using additional candidate genes namely *IL-6* and *IL-15* to support *CypA* as internal control gene.

As far as *NRF2* is concerned, multiple studies have shown that expression of *NRF2* is suppressed/ minimally induced in COVID-19 and that regulation of *NRF2* expression by SARS-CoV-2 is a viral strategy to hijack host protective responses (Zhang *et al.*, 2022). We first show that our data on *NRF2* expression based on *CypA*-normalization is actually consistent with the previous reports on *NRF2* expression showing that SARS-CoV-2 infection dysregulates *NRF2* to impair its proper induction leading to an increased oxidative stress, a hallmark feature of COVID-19 (Olagnier *et al.*, 2020; Zhang *et al.*, 2022). Similar to RNA sequencing-based studies, normalization with *CypA* revealed a relatively low *NRF2* expression even in the oxidative stress environment of COVID-19-associated inflammation.

Moreover, we did not observe significant difference in the expression of *NRF2* across the spectrum of COVID-19 infection (Figure 4, rebuttal and Fig. 4 in revised manuscript), as is also indicated from RNA Sequencing analysis (Jain *et al.*, 2021).

There is no published study involving CAM patients for us to correlate or compare *NRF2* expression. Slightly higher *NRF2* levels observed in CAM group in our study can be a response to mucormycosis infection, and *NRF2* levels returns closer to base-line following reduction in fungal burden post-surgery (post-CAM) (Figure 4 of rebuttal, and Fig. 4 in revised manuscript). Nonetheless, further focused studies would be required to study *NRF2* and redox regulation in CAM patients which is beyond the scope of this work.

We have incorporated these comments in the revised manuscript.

Minor comments

Comment 4: The official name of SARS-CoV-2 disease is COVID-19.

Response: We regret the oversight and this have been fixed in the revised manuscript.

Comment 5: Figure 1 and Figure 2 are distorted.

Response: This has now been fixed.

Comment 6: Line 223-224: "significant up-regulation of NRF2 was observed in mild (>250 fold), asymptomatic (>15 fold), moderate (>12 fold)". The authors should specify which disease and the figure number related to these data.

Response: We regret this oversight. As suggested, we have now added related information in revised manuscript.

Comment 7: Line 110: change SARS CoV-2 by SARS-CoV-2

Response: We regret the oversight and this have been fixed in the revised manuscript.

Comment 8: Line 152-155: The authors should specify the figure number related to the data (Figure 3e).

Response: We apologise for this confusion and this have been fixed in the revised manuscript.

Comment 3Line 214-215: The authors should specify the table number related to the data (Table 2).

Response: We apologise for this confusion and this have been fixed in the revised manuscript.

Taken together, in the revised work, we have added significant amount of new data including gene expression and protein estimation to adequately justify the relevance of *CypA* as the most suitable gene for qRT-PCR based gene expression analysis.

We thank both the reviewers for valuable comments and constructive suggestions that has allowed us to improve the manuscript significantly.

References:

- Banerjee AK, Blanco MR, Bruce EA, Honson DD, Chen LM, Chow A, Bhat P, Ollikainen N, Quinodoz SA, Loney C (2020) SARS-CoV-2 disrupts splicing, translation, and protein trafficking to suppress host defenses. *Cell* 183: 1325-1339. e1321
- Banki L, Büki A, Horvath G, Kekesi G, Kis G, Somogyvári F, Jancsó G, Vécsei L, Varga E, Tuboly G (2020) Distinct changes in chronic pain sensitivity and oxytocin receptor expression in a new rat model (Wisket) of schizophrenia. *Neuroscience Letters* 714: 134561
- Brym P, Ruśc A, Kamiński S (2013) Evaluation of reference genes for qRT-PCR gene expression studies in whole blood samples from healthy and leukemia-virus infected cattle. *Veterinary immunology and immunopathology* 153: 302-307
- Delgado-Roche, L., & Mesta, F. (2020). Oxidative stress as key player in severe acute respiratory syndrome coronavirus (SARS-CoV) infection. *Archives of medical research* 51(5): 384-387.
- Dheda K, Huggett JF, Bustin SA, Johnson MA, Rook G, Zumla A (2004) Validation of housekeeping genes for normalizing RNA expression in real-time PCR. *Biotechniques* 37: 112-119
- Finkel Y, Gluck A, Nachshon A, Winkler R, Fisher T, Rozman B, Mizrahi O, Lubelsky Y, Zuckerman B, Slobodin B (2021) SARS-CoV-2 uses a multipronged strategy to impede host protein synthesis. *Nature* 594: 240-245
- Fisher T, Gluck A, Narayanan K, Kuroda M, Nachshon A, Hsu JC, Halfmann PJ, Yahalom-Ronen Y, Tamir H, Finkel Y (2022) Parsing the role of NSP1 in SARS-CoV-2 infection. *Cell reports* 39: 110954
- Gao Q, Wang X-Y, Fan J, Qiu S-J, Zhou J, Shi Y-H, Xiao Y-S, Xu Y, Huang X-W, Sun J (2008) Selection of reference genes for real-time PCR in human hepatocellular carcinoma tissues. *Journal of cancer research and clinical oncology* 134: 979-986
- Ghawana S, Paul A, Kumar H, Kumar A, Singh H, Bhardwaj PK, Rani A, Singh RS, Raizada J, Singh K (2011) An RNA isolation system for plant tissues rich in secondary metabolites. *BMC research notes* 4: 1-5
- Islam, A. B. M. M., Khan, M., Ahmed, R., Hossain, M., Kabir, S. M., Islam, M., & Siddiki, A. M. A. M. (2021). Transcriptome of nasopharyngeal samples from COVID-19 patients and a comparative analysis with other SARS-CoV-2 infection models reveal disparate host responses against SARS-CoV-2. *Journal of Translational Medicine* 19(1): 1-25.
- Jain, R., Ramaswamy, S., Harilal, D., Uddin, M., Loney, T., Nowotny, N., ... & Abou Tayoun, A. (2021). Host transcriptomic profiling of COVID-19 patients with mild, moderate,

and severe clinical outcomes. *Computational and structural biotechnology journal* 19: 153-160.

Kidd M, Nadler B, Mane S, Eick G, Malfertheiner M, Champaneria M, Pfragner R, Modlin I (2007) GeneChip, geNorm, and gastrointestinal tumors: novel reference genes for real-time PCR. *Physiological genomics* 30: 363-370

Kobayashi EH, Suzuki T, Funayama R, Nagashima T, Hayashi M, Sekine H, Tanaka N, Moriguchi T, Motohashi H, Nakayama K (2016) Nrf2 suppresses macrophage inflammatory response by blocking proinflammatory cytokine transcription. *Nature communications* 7: 1-14

Kuang J, Yan X, Genders AJ, Granata C, Bishop DJ (2018) An overview of technical considerations when using quantitative real-time PCR analysis of gene expression in human exercise research. *PloS one* 13: e0196438

Liu Y, Qiu Y, Chen Q, Han X, Cai M, Hao L (2021) Puerarin suppresses the hepatic gluconeogenesis via activation of PI3K/Akt signaling pathway in diabetic rats and HepG2 cells. *Biomedicine & Pharmacotherapy* 137: 111325

Matsuoka Y, Nakayama H, Yoshida R, Hirose A, Nagata M, Tanaka T, Kawahara K, Sakata J, Arita H, Nakashima H (2016) IL-6 controls resistance to radiation by suppressing oxidative stress via the Nrf2-antioxidant pathway in oral squamous cell carcinoma. *British journal of cancer* 115: 1234-1244

McCord JM, Hybertson BM, Cota-Gomez A, Gao B (2021) Nrf2 activator PB125® as a carnosic acid-based therapeutic agent against respiratory viral diseases, including COVID-19. *Free Radical Biology and Medicine* 175: 56-64

Mehta R, Birerdinc A, Hossain N, Afendy A, Chandhoke V, Younossi Z, Baranova A (2010) Validation of endogenous reference genes for qRT-PCR analysis of human visceral adipose samples. *BMC Molecular Biology* 11: 1-10

Millet P, Vachharajani V, McPhail L, Yoza B, McCall CE (2016) GAPDH binding to TNF- α mRNA contributes to posttranscriptional repression in monocytes: a novel mechanism of communication between inflammation and metabolism. *The Journal of Immunology* 196: 2541-2551

Montero-Melendez T, Perretti M (2014) Gapdh gene expression is modulated by inflammatory arthritis and is not suitable for qPCR normalization. *Inflammation* 37: 1059-1069

Mukhopadhyay, S., Sinha, S., & Mohapatra, S. K. (2021). Analysis of transcriptomic data sets supports the role of IL-6 in NETosis and immunothrombosis in severe COVID-19. *BMC Genomic Data* 22(1): 1-14.

Okamoto T, Okabe S (2000) Ultraviolet absorbance at 260 and 280 nm in RNA measurement is dependent on measurement solution. *International journal of molecular medicine* 5: 657-666

- Olagnier D, Farahani E, Thyrssted J, Blay-Cadanet J, Herengt A, Idorn M, Hait A, Hernaez B, Knudsen A, Iversen MB (2020) SARS-CoV2-mediated suppression of NRF2-signaling reveals potent antiviral and anti-inflammatory activity of 4-octyl-itaconate and dimethyl fumarate. *Nature communications* 11: 1-12
- Pilbrow AP, Ellmers LJ, Black MA, Moravec CS, Sweet WE, Troughton RW, Richards AM, Frampton CM, Cameron VA (2008) Genomic selection of reference genes for real-time PCR in human myocardium. *BMC medical genomics* 1: 1-12
- Radonić A, Thulke S, Mackay IM, Landt O, Siegert W, Nitsche A (2004) Guideline to reference gene selection for quantitative real-time PCR. *Biochemical and biophysical research communications* 313: 856-862
- Seres-Bokor A, Kemény KK, Taherigorji H, Schaffer A, Kothencz A, Gáspár R, Ducza E (2021) The Effect of Citral on Aquaporin 5 and Trpv4 Expressions and Uterine Contraction in Rat—An Alternative Mechanism. *Life* 11: 897
- Shivashankar G, Lim JC, Acosta ML (2021) Glyceraldehyde-3-phosphate dehydrogenase and glutamine synthetase inhibition in the presence of pro-inflammatory cytokines contribute to the metabolic imbalance of diabetic retinopathy. *Experimental Eye Research* 213: 108845
- Silver N, Best S, Jiang J, Thein SL (2006) Selection of housekeeping genes for gene expression studies in human reticulocytes using real-time PCR. *BMC molecular biology* 7: 1-9
- Tantoh DM, Wu M-F, Ho C-C, Lung C-C, Lee K-J, Nfor ON, Liaw Y-C, Hsu S-Y, Chen P-H, Lin C (2019) SOX2 promoter hypermethylation in non-smoking Taiwanese adults residing in air pollution areas. *Clinical epigenetics* 11: 46
- Willinger CM, Rong J, Tanriverdi K, Courchesne PL, Huan T, Wasserman GA, Lin H, Dupuis J, Joehanes R, Jones MR (2017) MicroRNA signature of cigarette smoking and evidence for a putative causal role of microRNAs in smoking-related inflammation and target organ damage. *Circulation: Cardiovascular Genetics* 10: e001678
- Yu YJ, Majumdar AP, Nechvatal JM, Ram JL, Basson MD, Heilbrun LK, Kato I (2008) Exfoliated cells in stool: A source for Reverse Transcription-PCR-based analysis of biomarkers of gastrointestinal cancer. *Cancer Epidemiology Biomarkers & Prevention* 17: 455-458
- Zhang L, Yang F (2022) Tanshinone IIA improves diabetes-induced renal fibrosis by regulating the miR-34-5p/Notch1 axis. *Food Science & Nutrition*
- Zhang S, Wang J, Wang L, Aliyari S, Cheng G (2022) SARS-CoV-2 virus NSP14 Impairs NRF2/HMOX1 activation by targeting Sirtuin 1. *Cellular & molecular immunology*: 1-11

October 17, 2022

Dr. Vikram Saini
All India Institute of Medical Sciences, New Delhi
Biotechnology
New Delhi
India

Re: Spectrum01656-22R1 (Selection of ideal reference genes for gene expression analysis in COVID-19, and Mucormycosis infection)

Dear Dr. Vikram Saini:

Thank you for submitting your manuscript to Microbiology Spectrum. Your manuscript has been accepted, and I am forwarding it to the ASM Journals Department for publication. You will be notified when your proofs are ready to be viewed.

Sincerely,

Zsolt Toth
Editor, Microbiology Spectrum

Comments

Authors have addressed all the comments and I am convinced with their answers. They have performed additional experiments and revised the text as suggested. Overall, the revised manuscript is improved.